# GIT RE-BASIN: MERGING MODELS MODULO PERMUTATION SYMMETRIES

**Samuel K. Ainsworth, Jonathan Hayase, Siddhartha Srinivasa**
Paul G. Allen School of Computer Science and Engineering
University of Washington
`{skainswo,jhayase,siddh}@cs.washington.edu`

## ABSTRACT

The success of deep learning is due in large part to our ability to solve certain massive non-convex optimization problems with relative ease. Though non-convex optimization is NP-hard, simple algorithms – often variants of stochastic gradient descent – exhibit surprising effectiveness in fitting large neural networks in practice. We argue that neural network loss landscapes often contain (nearly) a single basin after accounting for all possible permutation symmetries of hidden units a la Entezari et al. (2021). We introduce three algorithms to permute the units of one model to bring them into alignment with a reference model in order to merge the two models in weight space. This transformation produces a functionally equivalent set of weights that lie in an approximately convex basin near the reference model. Experimentally, we demonstrate the single basin phenomenon across a variety of model architectures and datasets, including the first (to our knowledge) demonstration of zero-barrier linear mode connectivity between independently trained ResNet models on CIFAR-10. Additionally, we investigate intriguing phenomena relating model width and training time to mode connectivity. Finally, we discuss shortcomings of the linear mode connectivity hypothesis, including a counterexample to the single basin theory.

## 1 INTRODUCTION

We investigate the unreasonable effectiveness of stochastic gradient descent (SGD) algorithms on the high-dimensional non-convex optimization problems of deep learning. In particular,

1. Why does SGD thrive in optimizing high-dimensional non-convex deep learning loss landscapes despite being noticeably less robust in other non-convex optimization settings, like policy learning (Ainsworth et al., 2021), trajectory optimization (Kelly, 2017), and recommender systems (Kang et al., 2016)?

2. What are all the local minima? When linearly interpolating between initialization and final trained weights, why does the loss smoothly and monotonically decrease (Goodfellow & Vinyals, 2015; Frankle, 2020; Lucas et al., 2021; Vlaar & Frankle, 2021)?

3. How can two independently trained models with different random initializations and data batch orders inevitably achieve nearly identical performance? Furthermore, why do their training loss curves often look identical?

We posit that these phenomena point to the existence of some yet uncharacterized invariance(s) in the training dynamics causing independent training runs to exhibit similar characteristics. Hecht-Nielsen (1990) noted the permutation symmetries of hidden units in neural networks; briefly, one can swap any two units of a hidden layer in a network and – assuming weights are adjusted accordingly – network functionality will not change. Recently, Benton et al. (2021) demonstrated that SGD solutions form a connected volume of low loss and Entezari et al. (2021) conjectured that this volume is convex modulo permutation symmetries.

**Conjecture 1** (Permutation invariance, informal (Entezari et al., 2021))**.** Most SGD solutions belong to a set whose elements can be permuted so that no barrier (as in Definition 2.2) exists on the linear interpolation between any two permuted elements.

| ARCHITECTURE | NUM. PERMUTATION SYMMETRIES |
|---|---|
| MLP (3 layers, 512 width) | $10 \wedge 3498$ |
| VGG16 | $10 \wedge 35160$ |
| ResNet50 | $10 \wedge 55109$ |
| Atoms in the observable universe | $10 \wedge 82$ |

Table 1: **Permutation symmetries of deep learning models vs. an upper estimate on the number of atoms in the known, observable universe.** Deep learning loss landscapes contain incomprehensible amounts of geometric repetition.

We refer to such solutions as being *linearly mode connected* (LMC) (Frankle et al., 2020), an extension of mode connectivity (Garipov et al., 2018; Draxler et al., 2018). If true, Conjecture 1 will both materially expand our understanding of how SGD works in the context of deep learning and offer a credible explanation for the preceding phenomena, in particular.

**Contributions.** In this paper, we attempt to uncover what invariances may be responsible for the phenomena cited above and the unreasonable effectiveness of SGD in deep learning. We make the following contributions:

1. **Matching methods.** We propose three algorithms, grounded in concepts and techniques from combinatorial optimization, to align the weights of two independently trained models. Where appropriate, we prove hardness results for these problems and propose approximation algorithms. Our fastest method identifies permutations in mere seconds on current hardware.

2. **Relationship to optimization algorithms.** We demonstrate by means of counterexample that linear mode connectivity is an emergent property of training procedures, not of model architectures. We connect this result to prior work on the implicit biases of SGD.

3. **Experiments, including zero-barrier LMC for ResNets.** Empirically, we explore the existence of linear mode connectivity modulo permutation symmetries in experiments across MLPs, CNNs, and ResNets trained on MNIST, CIFAR-10, and CIFAR-100. We contribute the first-ever demonstration of zero-barrier LMC between two independently trained ResNets. We explore the relationship between LMC and model width as well as training time. Finally, we show evidence of our methods' ability to combine models trained on independent datasets into a merged model that outperforms both input models in terms of test loss (but not accuracy) and is no more expensive in compute or memory than either input model.

## 2 BACKGROUND

Although our methods can be applied to arbitrary model architectures, we proceed with the multi-layer perceptron (MLP) for its ease of presentation (Bishop, 2007). Consider an $L$-layer MLP,

$$f(\boldsymbol{x}; \Theta) = \boldsymbol{z}_{L+1}, \quad \boldsymbol{z}_{\ell+1} = \sigma(\boldsymbol{W}_\ell \boldsymbol{z}_\ell + \boldsymbol{b}_\ell), \quad \boldsymbol{z}_1 = \boldsymbol{x},$$

where $\sigma$ denotes an element-wise nonlinear activation function. Furthermore, consider a loss, $\mathcal{L}(\Theta)$, that measures the suitability of a particular set of weights $\Theta$ towards some goal, e.g., fitting to a training dataset.

Central to our investigation is the phenomenon of *permutation symmetries* of weight space. Given $\Theta$, we can apply some permutation to the output features of any intermediate layer, $\ell$, of the model, denoted by a permutation matrix $\boldsymbol{P} \in S_d$,[1]

$$\boldsymbol{z}_{\ell+1} = \boldsymbol{P}^\top \boldsymbol{P} \boldsymbol{z}_{\ell+1} = \boldsymbol{P}^\top \boldsymbol{P} \sigma(\boldsymbol{W}_\ell \boldsymbol{z}_\ell + \boldsymbol{b}_\ell) = \boldsymbol{P}^\top \sigma(\boldsymbol{P} \boldsymbol{W}_\ell \boldsymbol{z}_\ell + \boldsymbol{P} \boldsymbol{b}_\ell)$$

for $\sigma$, an element-wise operator. It follows that as long as we reorder the input weights of layer $\ell+1$ according to $\boldsymbol{P}^\top$, we will have a functionally equivalent model. To be precise, if we define $\Theta'$ to be identical to $\Theta$ with the exception of

$$\boldsymbol{W}_\ell' = \boldsymbol{P} \boldsymbol{W}_\ell, \quad \boldsymbol{b}_\ell' = \boldsymbol{P} \boldsymbol{b}_\ell, \quad \boldsymbol{W}_{\ell+1}' = \boldsymbol{W}_{\ell+1} \boldsymbol{P}^\top,$$

---

[1]We denote the set of all $d \times d$ permutation matrices – isomorphic to the symmetric group – as $S_d$, to the possible chagrin of pure mathematicians.

then the two models are functionally equivalent: $f(\boldsymbol{x}; \Theta) = f(\boldsymbol{x}; \Theta')$ for all inputs $\boldsymbol{x}$. This implies that for any trained weights $\Theta$, there is an entire equivalence class of functionally equivalent weight assignments, not just one such $\Theta$, and convergence to any one specific element of this equivalence class, as opposed to any others, is determined only by random seed. We denote a functionality-preserving permutation of weights as $\pi(\Theta)$.

Consider the task of reconciling the weights of two, independently trained models, $A$ and $B$, with weights $\Theta_A$ and $\Theta_B$, respectively, such that we can linearly interpolate between them. We assume that models $A$ and $B$ were trained with equivalent architectures but different random initializations, data orders, and potentially different hyperparameters or datasets, as well. Our central question is: Given $\Theta_A$ and $\Theta_B$, can we identify some $\pi$ such that when linearly interpolating between $\Theta_A$ and $\pi(\Theta_B)$, all intermediate models enjoy performance similar to $\Theta_A$ and $\Theta_B$?

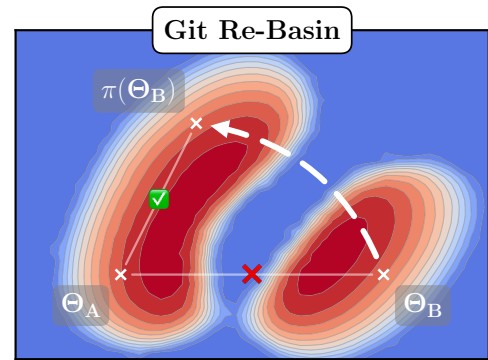

Figure 1: **Git Re-Basin merges models by teleporting solutions into a single basin.** $\Theta_B$ is permuted into functionally-equivalent $\pi(\Theta_B)$ so that it lies in the same basin as $\Theta_A$.

We base any claims of loss landscape convexity on the usual definition of multi-dimensional convexity in terms of one-dimensional convexity per

**Definition 2.1** (Convexity). A function $f : \mathbb{R}^D \to \mathbb{R}$ is convex if every one-dimensional slice is convex, i.e., for all $x, y \in \mathbb{R}^D$, the function $g(\lambda) = f((1 - \lambda)x + \lambda y)$ is convex in $\lambda$.

Due to Definition 2.1, it suffices to show that arbitrary one-dimensional slices of a function are convex in order to reason about the convexity of complex, high-dimensional functions. In practice, we rarely observe perfect convexity but instead hope to approximate it as closely as possible. Following Frankle et al. (2020); Entezari et al. (2021); Draxler et al. (2018); Garipov et al. (2018) and others, we measure approximations to convexity via "barriers."

**Definition 2.2** (Loss barrier (Frankle et al., 2020)). Given two points $\Theta_A, \Theta_B$ such that $\mathcal{L}(\Theta_A) \approx \mathcal{L}(\Theta_B)$, the *loss barrier* is defined as $\max_{\lambda \in [0,1]} \mathcal{L}((1 - \lambda)\Theta_A + \lambda \Theta_B) - \frac{1}{2}(\mathcal{L}(\Theta_A) + \mathcal{L}(\Theta_B))$.

Loss barriers are non-negative, with zero indicating an interpolation of flat or positive curvature.

## 3 PERMUTATION SELECTION METHODS

We introduce three methods of matching units between model $A$ and model $B$. Further, we present an extension to simultaneously merging multiple models in Appendix A.10 and an appealing but failed method in Appendix A.11.

### 3.1 MATCHING ACTIVATIONS

Following the classic Hebbian mantra, "[neural network units] that fire together, wire together" (Hebb, 2005), we consider associating units across two models by performing regression between their activations. Matching activations between models is compelling since it captures the intuitive notion that two models must learn similar features to accomplish the same task (Li et al., 2016). Provided activations for each model, we aim to associate each unit in $A$ with a unit in $B$. It stands to reason that a linear relationship may exist between the activations of the two models. We fit this into the regression framework by constraining ordinary least squares (OLS) to solutions in the set of permutation matrices, $S_d$. For activations of the $\ell$'th layer, let $\boldsymbol{Z}^{(A)}, \boldsymbol{Z}^{(B)} \in \mathbb{R}^{d \times n}$ denote the $d$-dim. activations for all $n$ training data points in models $A$ and $B$, respectively. Then,

$$\boldsymbol{P}_\ell = \underset{\boldsymbol{P} \in S_d}{\arg\min} \sum_{i=1}^{n} \|\boldsymbol{Z}^{(A)}_{:,i} - \boldsymbol{P}\boldsymbol{Z}^{(B)}_{:,i}\|^2 = \underset{\boldsymbol{P} \in S_d}{\arg\max} \langle \boldsymbol{P}, \boldsymbol{Z}^{(A)}(\boldsymbol{Z}^{(B)})^\top \rangle_F, \qquad (1)$$

where $\langle \boldsymbol{A}, \boldsymbol{B} \rangle_F = \sum_{i,j} A_{i,j} B_{i,j}$ denotes the Frobenius inner product between real-valued matrices $\boldsymbol{A}$ and $\boldsymbol{B}$. Conveniently, eq. (1) constitutes a "linear assignment problem" (LAP) (Bertsekas, 1998) for which efficient, practical algorithms are known. Having solved this assignment problem on each layer, we can then permute the weights of model $B$ to match model $A$ as closely as possible

$$\boldsymbol{W}_\ell' = \boldsymbol{P}_\ell \boldsymbol{W}_\ell^{(B)} \boldsymbol{P}_{\ell-1}^\top, \quad \boldsymbol{b}_\ell' = \boldsymbol{P}_\ell \boldsymbol{b}_\ell^{(B)}$$

for each layer $\ell$, producing weights $\Theta'$ with activations that align as closely possible with $\Theta_A$.

Computationally, this entire process is relatively lightweight: the $\boldsymbol{Z}^{(A)}$ and $\boldsymbol{Z}^{(B)}$ matrices can be computed in a single pass over the training dataset, and, in practice, a full run through the training dataset may be unnecessary. Solving eq. (1) is possible due to well-established, polynomial-time algorithms for solving the linear assignment problem (Kuhn, 2010; Jonker & Volgenant, 1987; Crouse, 2016). Also, conveniently, the activation matching at each layer is independent of the matching at every other layer, resulting in a separable and straightforward optimization problem; this advantage will not be enjoyed by the following methods.

Dispensing with regression, one could similarly associate units by matching against a matrix of cross-correlation coefficients in place of $\boldsymbol{Z}^{(A)}(\boldsymbol{Z}^{(B)})^\top$. We observed correlation matching to work equally well but found OLS regression matching to be more principled and easier to implement.

Activation matching has previously been studied for model merging in Tatro et al. (2020); Singh & Jaggi (2020); Li et al. (2016) albeit not from the perspective of OLS regression.

## 3.2 Matching Weights

Instead of associating units by their activations, we could alternatively inspect the weights of the model itself. Consider the first layer weights, $\boldsymbol{W}_1$; each row of $\boldsymbol{W}_1$ corresponds to a single feature. If two such rows were equal, they would compute exactly the same feature (ignoring bias terms for the time being). And, if $[\boldsymbol{W}_1^{(A)}]_{i,:} \approx [\boldsymbol{W}_1^{(B)}]_{j,:}$, it stands to reason that units $i$ and $j$ should be associated. Extending this idea to every layer, we are inspired to pursue the optimization

$$\arg\min_\pi \ \|\mathrm{vec}(\Theta_A) - \mathrm{vec}(\pi(\Theta_B))\|^2 \ = \ \arg\max_\pi \ \mathrm{vec}(\Theta_A) \cdot \mathrm{vec}(\pi(\Theta_B)).$$

We can re-express this in terms of the full weights,

$$\arg\max_{\pi=\{\boldsymbol{P}_i\}} \langle \boldsymbol{W}_1^{(A)}, \ \boldsymbol{P}_1 \boldsymbol{W}_1^{(B)} \rangle_F + \langle \boldsymbol{W}_2^{(A)}, \ \boldsymbol{P}_2 \boldsymbol{W}_2^{(B)} \boldsymbol{P}_1^\top \rangle_F + \cdots + \langle \boldsymbol{W}_L^{(A)}, \ \boldsymbol{W}_L^{(B)} \boldsymbol{P}_{L-1}^\top \rangle_F, \quad (2)$$

resulting in another matching problem. We term this formulation the "sum of bilinear assignments problem" (SOBLAP). Unfortunately, this matching problem is thornier than the classic linear assignment matching problem presented in eq. (1). Unlike LAP, we are interested in permuting *both* the rows and columns of $\boldsymbol{W}_\ell^{(B)}$ to match $\boldsymbol{W}_\ell^{(A)}$, which fundamentally differs from permuting only rows or only columns. We formalize this difficulty as follows.

**Lemma 1.** *The sum of a bilinear assignments problem (SOBLAP) is NP-hard and admits no polynomial-time constant-factor approximation scheme for $L > 2$.*

Lemma 1 contrasts starkly with classical LAP, for which polynomial-time algorithms are known.

Undeterred, we propose a approximation algorithm for SOBLAP. Looking at a single $\boldsymbol{P}_\ell$ while holding the others fixed, we observe that the problem can be reduced to a classic LAP,

$$\arg\max_{\boldsymbol{P}_\ell} \langle \boldsymbol{W}_\ell^{(A)}, \ \boldsymbol{P}_\ell \boldsymbol{W}_\ell^{(B)} \boldsymbol{P}_{\ell-1}^\top \rangle_F + \langle \boldsymbol{W}_{\ell+1}^{(A)}, \ \boldsymbol{P}_{\ell+1} \boldsymbol{W}_{\ell+1}^{(B)} \boldsymbol{P}_\ell^\top \rangle_F$$

$$= \arg\max_{\boldsymbol{P}_\ell} \langle \boldsymbol{P}_\ell, \ \boldsymbol{W}_\ell^{(A)} \boldsymbol{P}_{\ell-1} (\boldsymbol{W}_\ell^{(B)})^\top + (\boldsymbol{W}_{\ell+1}^{(A)})^\top \boldsymbol{P}_{\ell+1} \boldsymbol{W}_{\ell+1}^{(B)} \rangle_F.$$

This leads to a convenient coordinate descent algorithm: go through each layer and greedily select its best $\boldsymbol{P}_\ell$. Repeat until convergence. We present this in Algorithm 1.

Although we present Algorithm 1 in terms of an MLP without bias terms, in practice our implementation can handle the weights of models of nearly arbitrary architectures, including bias terms, residual connections, convolutional layers, attention mechanisms, and so forth. We propose an extension of Algorithm 1 to merging more than two models at a time in Appendix A.10.

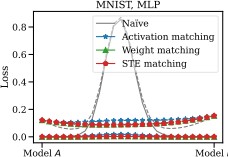 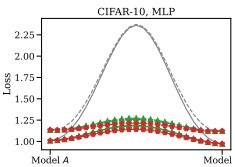 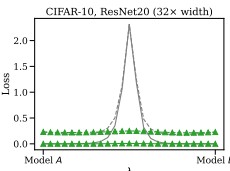 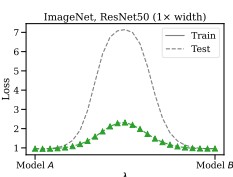

Figure 2: **Linear mode connectivity is possible after permuting.** Loss landscapes when interpolating between models trained on MNIST, CIFAR-10, and ImageNet. In all cases we can significantly improve over naïve interpolation. Straight-through estimator matching performs best but is very computationally expensive. Weight and activation matching perform similarly, although weight matching is orders of magnitude faster and does not rely on the input data distribution. We hypothesize that the ImageNet barrier could be reduced by increasing the model width as in Section 5.3.

---

**Algorithm 1:** PERMUTATIONCOORDINATEDESCENT

**Given:** Model weights $\Theta_A = \left\{ \boldsymbol{W}_1^{(A)}, \dots, \boldsymbol{W}_L^{(A)} \right\}$ and $\Theta_B = \left\{ \boldsymbol{W}_1^{(B)}, \dots, \boldsymbol{W}_L^{(B)} \right\}$

**Result:** A permutation $\pi = \{\boldsymbol{P}_1, \dots, \boldsymbol{P}_{L-1}\}$ of $\Theta_B$ such that $\mathrm{vec}(\Theta_A) \cdot \mathrm{vec}(\pi(\Theta_B))$ is approximately maximized.

---

**Initialize:** $\boldsymbol{P}_1 \leftarrow \boldsymbol{I}, \dots, \boldsymbol{P}_{L-1} \leftarrow \boldsymbol{I}$
**repeat**
    **for** $\ell \in$ RANDOMPERMUTATION$(1, \dots, L-1)$ **do**
        $\boldsymbol{P}_\ell \leftarrow$ SOLVELAP $\left( \boldsymbol{W}_\ell^{(A)} \boldsymbol{P}_{\ell-1} (\boldsymbol{W}_\ell^{(B)})^\top + (\boldsymbol{W}_{\ell+1}^{(A)})^\top \boldsymbol{P}_{\ell+1} \boldsymbol{W}_{\ell+1}^{(B)} \right)$
    **end**
**until** *convergence*

---

**Lemma 2.** *Algorithm 1 terminates.*

Our experiments showed this algorithm to be fast in terms of both iterations necessary for convergence and wall-clock time, generally on the order of seconds to a few minutes.

Unlike the activation matching method presented in Section 3.1, weight matching ignores the data distribution entirely. Ignoring the input data distribution and therefore the loss landscape handicaps weight matching but allows it to be much faster. We therefore anticipate its potential application in fields such as finetuning (Devlin et al., 2019; Wortsman et al., 2022b;a), federated learning (McMahan et al., 2017; Konečný et al., 2016a;b), and model patching (Matena & Raffel, 2021; Sung et al., 2021; Raffel, 2021). In practice, we found weight matching to be surprisingly competitive with data-aware methods. Section 5 studies this trade-off.

### 3.3 LEARNING PERMUTATIONS WITH A STRAIGHT-THROUGH ESTIMATOR

Inspired by the success of straight-through estimators (STEs) in other discrete optimization problems (Bengio et al., 2013; Kusupati et al., 2021; Rastegari et al., 2016; Courbariaux & Bengio, 2016), we attempt here to "learn" the ideal permutation of weights $\pi(\Theta_B)$. Specifically, our goal is to optimize

$$\min_{\tilde{\Theta}_B} \mathcal{L}\left( \frac{1}{2} \left( \Theta_A + \mathrm{proj}\left( \tilde{\Theta}_B \right) \right) \right), \qquad \mathrm{proj}(\Theta) \triangleq \arg\max_{\pi} \ \mathrm{vec}(\Theta) \cdot \mathrm{vec}(\pi(\Theta_B)), \qquad (3)$$

where $\tilde{\Theta}_B$ denotes an approximation of $\pi(\Theta_B)$, allowing us to implicitly optimize $\pi$. However, eq. (3) involves inconvenient non-differentiable projection operations, $\mathrm{proj}(\cdot)$, complicating the optimization. We overcome this via a "straight-through" estimator: we parameterize the problem in terms of a set of weights $\tilde{\Theta}_B \approx \pi(\Theta_B)$. In the forward pass, we project $\tilde{\Theta}_B$ to the closest realizable $\pi(\Theta_B)$. In the backwards pass, we then switch back to the unrestricted weights $\tilde{\Theta}_B$. In this way, we

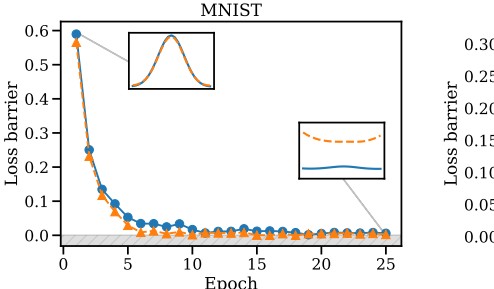 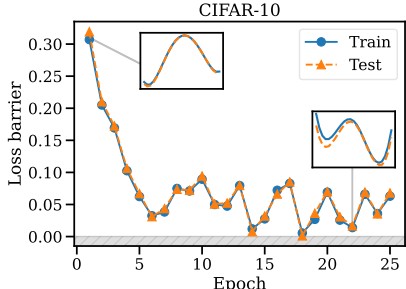

Figure 3: **Linear mode connectivity is challenging at initialization.** We show loss barriers per training time for MLPs trained on MNIST (left) and CIFAR-10 (right). Loss interpolation plots are inlaid to highlight results in initial and later epochs. LMC manifests gradually throughout training. We hypothesize that the variance in CIFAR-10 training is higher due to our MLP architecture being under-powered relative to the dataset. (Y-axis scales differ in each inlaid plot.)

are guaranteed to stay true to the projection constraints in evaluating the loss but can still compute usable gradients at our current parameters, $\tilde{\Theta}_B$.[2]

Conveniently, we can re-purpose Algorithm 1 to solve $\text{proj}(\tilde{\Theta}_B)$. Furthermore, we found that initializing $\tilde{\Theta}_B = \Theta_A$ performed better than random initialization. This is to be expected immediately at initialization since the initial matching will be equivalent to the weight matching method of Section 3.1. However, it is not immediately clear why these solutions continue to outperform a random initialization asymptotically.

Unlike the aforementioned methods, Algorithm 2 attempts to explicitly "learn" the best permutation $\pi$ using a conventional training loop. By initializing to the weight matching solution of Section 3.2 and leveraging the data distribution as in Section 3.1, it seeks to offer a best-of-both-worlds solution. However, this comes at a very steep computational cost relative to the other two methods.

## 4    A COUNTEREXAMPLE TO UNIVERSAL LINEAR MODE CONNECTIVITY

In this section we argue that common optimization algorithms, especially SGD and its relatives, are implicitly biased towards solutions admitting linear mode connectivity. In particular, we demonstrate – by way of a counterexample – that adversarial, non-SGD solutions exist in loss landscapes such that no permutation of units results in linear mode connectivity. We present this counterexample in complete detail in Appendix A.6.

The existence of adversarial basins suggests that our ability to find LMC between independently trained models is thanks to inherent biases in optimization methods. We emphasize that this counterexample does not contradict Conjecture 1; rather, it illustrates the importance of the conjecture's restriction to SGD solutions (Entezari et al., 2021). Characterizing the precise mechanism by which these solutions are biased towards LMC could be an exciting avenue for future work.

We also note that there are invariances beyond permutation symmetries: It is possible to move features between layers, re-scale layers, and so forth. Prior works noted the feature/layer association (Nguyen et al., 2021) and re-scaling invariances (Ainsworth et al., 2018). The importance of these other symmetries and their interplay with optimization algorithms remains unclear.

## 5    EXPERIMENTS

Our base methodology is to separately train two models, $A$ and $B$, starting from different random initializations and with different random batch orders, resulting in trained weights $\Theta_A$ and $\Theta_B$,

---

[2]Note again that projecting according to inner product distance is equivalent to projecting according to the $L_2$ distance when parameterizing the estimator based on the $B$ endpoint. We also experimented with learning the midpoint directly, $\tilde{\Theta} \approx \frac{1}{2}(\Theta_A + \pi(\Theta_B))$, in which case the $L_2$ and inner product projections diverge. In testing all possible variations, we found that optimizing the $B$ endpoint had a slight advantage, but all possible variations performed admirably.

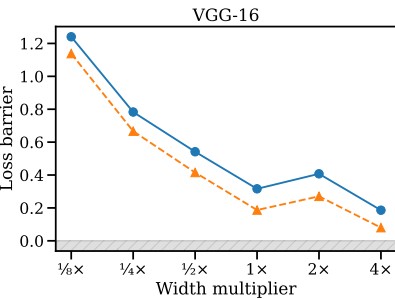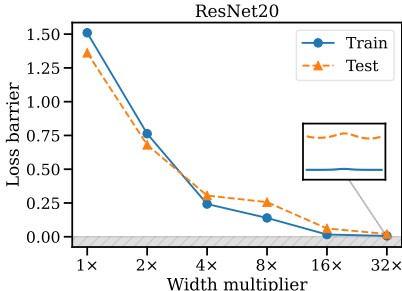

Figure 4: **Wider models exhibit better linear mode connectivity.** Training convolutional and ResNet architectures on CIFAR-10, we ablate their width and visualize loss barriers after weight matching. Notably, we achieve zero-barrier linear mode connectivity between ResNet models, the first such demonstration.

respectively. We then evaluate slices through the loss landscape, $\mathcal{L}((1 - \lambda)\Theta_A + \lambda\pi(\Theta_B))$ for $\lambda \in [0, 1]$, where $\pi$ is selected according to the methods presented in Section 3.[3] Ideally, we seek a completely flat or even convex one-dimensional slice. As discussed in Section 2, the ability to exhibit this behavior for arbitrary $\Theta_A, \Theta_B$ empirically suggests that the loss landscape contains only a single basin modulo permutation symmetries.

We remark that a failure to find a $\pi$ such that linear mode connectivity holds cannot rule out the existence of a satisfactory permutation. Given the astronomical number of permutation symmetries, Conjecture 1 is essentially impossible to disprove for any realistically wide model architecture.

## 5.1 Loss Landscapes Before and After Matching

We present results for models trained on MNIST (LeCun et al., 1998), CIFAR-10 (Krizhevsky, 2009), and ImageNet (Deng et al., 2009) in Figure 2. Naïve interpolation ($\pi(\Theta_B) = \Theta_B$) substantially degrades performance when interpolating. On the other hand, the methods introduced in Section 3 can achieve much better barriers. We achieve zero-barrier linear mode connectivity on MNIST with all three methods, although activation matching performs just slightly less favorably than weight matching and straight-through estimator (STE) matching. We especially note that the test loss landscape becomes convex after applying our weight matching and STE permutations! In other words, our interpolation actually yields a merged model that outperforms both models $A$ and $B$. We elaborate on this phenomenon in Section 5.4 and Appendix A.10.

On ImageNet we fall short of zero-barrier connections, although we do see a 67% decrease in barrier relative to naïve interpolation. As we demonstrate in Section 5.3, we can achieve zero-barrier LMC on CIFAR-10 with large ResNet models. Therefore, we hypothesize that the presence of LMC depends on the model having sufficient capacity (esp. width) to capture the complexity of the input data distribution, and that ImageNet results could be improved by expanding model width.

STE matching, the most expensive method, produces the best solutions. Somewhat surprising, however, is that the gap between STE and the other two methods is relatively small. In particular, it is remarkable how well Algorithm 1 performs without access to the input data at all. We found that weight matching offered a compelling balance between computational cost and performance: It runs in mere seconds (on current hardware) and produces high-quality solutions.

## 5.2 Onset of Mode Connectivity

Given the results of Section 5.1, it may be tempting to conclude that the entirety of weight space contains only a single basin modulo permutation symmetries. However, we found that linear mode connectivity is an emergent property of training, and we were unable to uncover it early in training. We explore the emergence of LMC in Figure 3. Concurrent to our work, Benzing et al. (2022) showed that LMC at initialization is possible using a permutation found at the end of training.

---

[3]We also experimented with spherical linear interpolation ("slerp") and found it to perform slightly better than linear interpolation in some cases; however, the difference was not sufficiently significant to warrant diverging from the pre-existing literature.

Note that the final inlaid interpolation plot in Figure 3(right) demonstrates an important shortcoming of the loss barrier metric, i.e., the interpolation includes points with lower loss than either of the two models. However, the loss barrier is still positive due to non-negativity, as mentioned in Section 2.

### 5.3 EFFECT OF MODEL WIDTH

Conventional wisdom maintains that wider architectures are easier to optimize (Jacot et al., 2018; Lee et al., 2019). We now investigate whether they are also easier to linearly mode connect. We train VGG-16 (Simonyan & Zisserman, 2015) and ResNet20 (He et al., 2016) architectures of varying widths on the CIFAR-10 dataset. Results are presented in Figure 4.[4]

A clear relationship emerges between model width and linear mode connectivity, as measured by the loss barrier between solutions. Although $1\times$-sized models did not seem to exhibit linear mode connectivity, we found that larger width models decreased loss barriers all the way to zero. In Figure 4(right), we show what is to our knowledge the premiere demonstration of zero-barrier linear mode connectivity between two large ResNet models trained on a non-trivial dataset.

We highlight that relatively thin models do not seem to obey linear mode connectivity yet still exhibit similarities in training dynamics. This suggests that either our permutation selection methods are failing to find satisfactory permutations on thinner models or that some form of invariance other than permutation symmetries must be at play in the thin model regime.

### 5.4 MODEL PATCHING, SPLIT DATA TRAINING, AND IMPROVED CALIBRATION

Inspired by work on finetuning (Wortsman et al., 2022a), model patching (Singh & Jaggi, 2020; Raffel, 2021), and federated learning (McMahan et al., 2017; Konečný et al., 2016a;b), we study whether it is possible to synergistically merge the weights of two models trained on disjoint datasets. Consider, for example, an organization with multiple (possibly biased) datasets separated for regulatory (e.g., GDPR) or privacy (e.g., on-device data) considerations. Models can be trained on each dataset individually, but training in aggregate is not feasible. Can we combine separately trained models so that the merged model performs well on the entirety of the data?

To address this question, we split the CIFAR-100 dataset (Krizhevsky, 2009) into two disjoint subsets: dataset $A$, containing 20% examples labelled 0-49 and 80% labelled 50-99, and dataset $B$, vice versa. ResNet20 models $A$ and

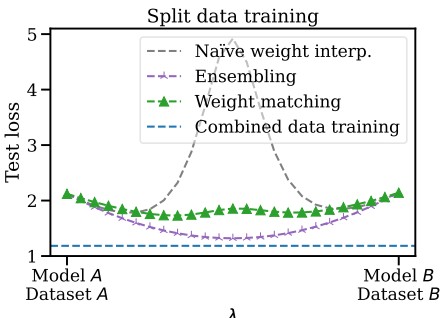

Figure 5: **Models trained on disjoint datasets can be merged constructively.** Algorithm 1 makes it possible for two ResNet models trained on disjoint, biased subsets of CIFAR-100 to be merged in weight space such that their combination outperforms both input models in terms of test loss on the combined dataset.

$B$ were trained on their corresponding datasets. Privacy requirements mandate that we utilize a data-agnostic algorithm like Algorithm 1. Figure 5 shows the result of merging the two models with weight matching. For comparison, we benchmark naïve weight interpolation, ensembling of the model logits, and full-data training.

As expected, merging separately trained models did not match the performance of an omniscient model trained on the full dataset or an ensemble of the two models with twice the number of effective weights. On the other hand, we did manage to merge the two models in weight space, achieving an interpolated model that outperforms both input models in terms of test loss while using half the memory and compute required for ensembling. Furthermore, the merged model's probability estimates are better calibrated than either of the input models as demonstrated in Figure 11. Accuracy results are presented in Figure 10. Algorithm 1 also vastly outperformed naïve interpolation, the status quo for model combination in federated learning and distributed training.

---

[4]Unfortunately, $8\times$ width VGG-16 training was unattainable since it exhausted GPU memory on available hardware at the time of writing.

## 6 RELATED WORK

**(Linear) mode connectivity.** Garipov et al. (2018); Draxler et al. (2018); Freeman & Bruna (2017) showed that different solutions in the neural network loss landscape could be connected by paths of near-constant loss, which Garipov et al. (2018) coined "mode connectivity." Tatro et al. (2020) explored non-linear mode connectivity modulo permutation symmetries. Frankle et al. (2020) demonstrated a connection between *linear* mode connectivity and the lottery ticket hypothesis. Juneja et al. (2022) demonstrated that LMC does not always hold, even when fine-tuning. Hecht-Nielsen (1990); Chen et al. (1993) noted the existence of permutation symmetries, and Brea et al. (2019) implicated them as a source of saddle points in the loss landscape. Recently, the prescient work of Entezari et al. (2021) conjectured that SGD solutions could be linear mode connected modulo permutation symmetries and offered experiments buttressing this claim. Unlike previous works on LMC we accomplish zero-barrier paths between two independently-trained models with an algorithm that runs on the order of seconds.

**Loss landscapes and training dynamics.** Li et al. (2016); Yosinski et al. (2014) investigated whether independently trained networks learn similar features, and to what extent they transfer. Jiang et al. (2021) argued that independently trained networks meaningfully differ in the features they learn in certain scenarios. Zhang et al. (2019) studied the relative importance of layers. Benton et al. (2021) argued that SGD solutions form a connected volume of low loss. Pittorino et al. (2022) proposed a toroidal topology of solutions and a set of algorithms for symmetry removal. On the theoretical front, Kawaguchi (2016) proved that deep linear networks contain no local minima. Boursier et al. (2022); Chizat & Bach (2018); Mei et al. (2018) characterized the training dynamics of one-hidden layer networks, proving that they converge to zero loss. Godfrey et al. (2022); Simsek et al. (2021) investigated the algebraic structure of symmetries in neural networks and how this structure manifests in loss landscape geometry.

**Federated learning and model merging.** McMahan et al. (2017); Konečný et al. (2016a;b) introduced the concept of "federated learning," i.e., learning split across across multiple devices and datasets. Wang et al. (2020) proposed an exciting federated learning method in which model averaging is done after permuting units. Unlike this work, they merged smaller "child" models into a larger "main" model, and did so with a layer-wise algorithm that does not support residual connections or normalization layers. Raffel (2021); Matena & Raffel (2021); Sung et al. (2021) conceptualized the study of "model patching," i.e., the idea that models should be easy to modify and submit changes to. Ilharco et al. (2022) investigated model patching for the fine-tuning of open-vocabulary vision models. Ashmore & Gashler (2015) first explored the use of matching algorithms for the alignment of network units. Singh & Jaggi (2020) proposed merging models by soft-aligning associations weights, inspired by optimal transport. Liu et al. (2022a); Uriot & Izzo (2020) further explored merging models taking possible permutations into account. Wortsman et al. (2022a) demonstrated state-of-the-art ImageNet performance by averaging the weights of many fine-tuned models.

## 7 DISCUSSION AND FUTURE WORK

We explore the role of permutation symmetries in the linear mode connectivity of SGD solutions. We present three algorithms to canonicalize independent neural network weights in order to make the loss landscape between them as flat as possible. In contrast to prior work, we linearly mode connect large ResNet models with no barrier in seconds to minutes. Despite presenting successes across multiple architectures and datasets, linear mode connectivity between thin models remains elusive. Therefore, we conjecture that permutation symmetries are a necessary piece, though not a complete picture, of the fundamental invariances at play in neural network training dynamics. In particular, we hypothesize that linear, possibly non-permutation, relationships connect the layer-wise activations between models trained by SGD. In the infinite width limit, there exist satisfactory linear relationships that are also permutations.

An expanded theory and empirical exploration of other invariances – such as cross-layer scaling or general linear relationships between activations – presents an intriguing avenue for future work. Ultimately, we anticipate that a lucid understanding of loss landscape geometry will not only advance the theory of deep learning but will also promote the development of better optimization, federated learning, and ensembling techniques.

ETHICS STATEMENT

Merging models raises interesting ethical and technical questions about the resulting models. Do they inherit the same biases as their input models? Are rare examples forgotten when merging? Is it possible to gerrymander a subset of the dataset by splitting its elements across many shards?

Deployment of any form of model merging ought to be paired with thorough auditing of the resulting model, investigating in particular whether the merged model is representative of the entirety of the data distribution.

REPRODUCIBILITY STATEMENT

Our code is open sourced at `https://github.com/samuela/git-re-basin`. Our experimental logs and downloadable model checkpoints are fully open source at `https://wandb.ai/skainswo/git-re-basin`.

ACKNOWLEDGMENTS

We are grateful to Vivek Ramanujan, Mitchell Wortsman, Aditya Kusupati, Rahim Entezari, Jason Yosinski, Krishna Pillutla, Ofir Press, Matt Wallingford, Tim Dettmers, Raghav Somani, Gabriel Ilharco, Ludwig Schmidt, Sewoong Oh, and Kevin Jamieson for enlightening discussions. Thank you to John Thickstun and Sandy Kaplan for their thoughtful review of an earlier draft of this work and to Ofir Press and Tim Dettmers for their potent advice on framing and communicating this work. This work was (partially) funded by the National Science Foundation NRI (#2132848) & CHS (#2007011), DARPA RACER, the Office of Naval Research, Honda Research Institute, and Amazon. This work is supported in part by Microsoft and NSF grants DMS-2134012 and CCF-2019844 as a part of NSF Institute for Foundations of Machine Learning (IFML).

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

# A  APPENDIX

## A.1  KNOWN FAILURE MODES

We emphasize that none of the techniques presented in this paper are silver bullets. Here we list the failure cases that the authors are presently aware of,

1. Models of insufficient width
2. Models in the initial stages of training
3. VGGs on MNIST
4. MNIST MLPs trained with SGD and too low of a learning rate, or Adam and too high of a learning rate
5. ConvNeXt architectures (Liu et al., 2022b), which have surprisingly few permutation symmetries due to extensive use of depth-wise convolutions

Furthermore, we believe other failure modes certainly exist but have yet to be discovered.

We are excited by the prospect of future work investigating these failure modes and improving our understanding of when and why model merging modulo permutation symmetries is feasible.

## A.2  EXTENDED RELATED WORK

**Non-linear mode connectivity.** A flourishing set of literature exists studying non-linear mode connectivity, including but not limited to Garipov et al. (2018); Draxler et al. (2018); Kuditipudi et al. (2019). This insightful line of work is inspirational to our own, however we take a strictly linear approach to mode connectivity as in Frankle et al. (2020); Juneja et al. (2022). Restricting ourselves to linear trajectories comes with the advantage of having direct implications for a single-basin theory. However, it comes at the cost of a more challenging, discrete optimization problem. In particular, we found that – in contrast to non-linear mode connectivity – linear mode connectivity becomes drastically harder with smaller width models. Note additionally that most pre-existing mode connectivity work does not account for permutation symmetries of weight space, a linchpin element of our work. A notable exception to this trend can be found in Tatro et al. (2020).

To summarize: Freeman & Bruna (2017) introduced the notion of mode connectivity and proved that loss landscapes for single-hidden layer ReLU models contain only a single basin in the infinite width limit. Garipov et al. (2018) and Draxler et al. (2018) concurrently demonstrated that simple zero-barrier curves can be learned to connect the optima in weight space, thus reshaping our understanding of practical loss landscape geometries. Kuditipudi et al. (2019) proposes a theoretical explanation for the mode connectivity phenomenon. Benton et al. (2021) extends mode connectivity from one-dimensional paths to entire manifolds of low-loss, and show that these manifolds can be leveraged for state-of-the-art Bayesian ensembling of models.

**Relationship with Tatro et al. (2020).** The impact of permutation symmetries on the non-linear mode connectivity of models is considered in Tatro et al. (2020). In particular, they independently propose an algorithm more-or-less equivalent to Section 3.1 but use it in conjunction with learned non-linear mode connecting curves. In contrast, we show that linear mode connectivity can be achieved without the need for learning non-linear paths between the aligned weights. Our derivation of Section 3.1 from the principle of least-squares regression is novel, to the best of our knowledge.

**Relationship with Singh & Jaggi (2020).** Singh & Jaggi (2020) studies model merging with "soft matchings" between units from the perspective of optimal transport. We emphasize the following commonalities/differences with their work:

- We focus on linear mode connectivity modulo permutation symmetries and its implications for a single-basin theory. On the other hand, Singh & Jaggi (2020) emphasizes "soft" (ie., non-permutation) matching of units via optimal transport.
- The activation matching method of Singh & Jaggi (2020) reduces to that of Section 3.1 when the optimal transport regularization term is set to zero and the unit "importance" values are set to uniform across all units on all layers.

- Our weight matching and straight-through estimator methods solve for an alignment across all layers jointly. In contrast, Singh & Jaggi (2020) executes greedy, single-pass matching looking only at weight information from the immediately previous layer when selecting permutations. Singh & Jaggi (2020) suggests jointly solving for alignments as an avenue for future work.

- The "wts" method of Singh & Jaggi (2020) is not run on models including bias terms, skip connections, or normalization layers. In contrast, our Algorithm 1 works with models of nearly arbitrary architecture.

- Our weight matching method (Algorithm 1) outperforms the "wts" method of Singh & Jaggi (2020). See Appendix A.7 for more information.

- Singh & Jaggi (2020) introduces a method for merging multiple models simultaneously, but only demonstrates results on at most 8 models at a time and performs continued training after merging. In contrast, our Algorithm 3 has been shown to work with as many as 32 models at a time and does not require continued training after merging. Our analysis of the calibration of the resulting merged models has no parallel in Singh & Jaggi (2020).

**Relationship with Entezari et al. (2021).** Entezari et al. (2021) introduces the single-basin conjecture and provides the following evidence towards it:

- Entezari et al. (2021) provides a statistical test which fails to detect a difference in barrier statistics between independently trained models and random permutations of the same model (Fig. 5 of Entezari et al. (2021)). Our work provides stronger support for the conjecture in that we give methods that can directly "unscramble" these permutations, proving that LMC can be found (Figures 4 and 5).

  Entezari et al. (2021)'s experimental protocol does not provide evidence for linear mode connectivity. Rather, their experimental results suggest that barriers resulting from independent training look like the barriers resulting from random permutations. But this result is consistent with a world in which all solutions have barriers between them – both between members of the same permutation equivalence class and between solutions in separate equivalence classes! In other words, there may still exist multiple equivalence classes of solutions. In contrast, we provide concrete evidence for a single-basin theory by developing algorithms that directly place independent solutions into the same basin (Figure 1).

- Although Entezari et al. (2021)'s conjecture is an important intellectual ancestor to our work, their demonstration of linear mode connectivity is limited to a single hidden-layer MLP on MNIST (Fig. 2 of Entezari et al. (2021)). However, this result for single hidden-layer MLP models is preceded by Freeman & Bruna (2017); Uriot & Izzo (2020). On the other hand, we focus on larger models and datasets that are more closely aligned with models used in practice at the time of writing.

- Entezari et al. (2021) proposes a simulated annealing algorithm that yields modest reductions in barrier between independently trained models, yet requires multiple days to run.

  On the other hand, our weight matching algorithm (Algorithm 1) completely removes barriers between models for more challenging models and datasets (Figures 2 and 5), and runs in seconds (Appendix A.5). Moreover, our weight matching method does not require access to the training data, enabling its potential application in domains like federated learning and distributed training.

In short, the work of Entezari et al. (2021) first proposed the "single-basin" conjecture. Our work is the first (to the best of our knowledge) to demonstrate that linear mode connectivity can be achieved between large models independently trained on challenging datasets.

**Differentiating through permutations.** Akin to differentiable permutation learning, many prior works have studied differentiable sorting (Grover et al., 2019; Prillo & Eisenschlos, 2020; Cuturi et al., 2019; Petersen et al., 2022; 2021; Mena et al., 2018). Blondel et al. (2020) studied differentiable sorting and ranking with asymptotics that correspond to their non-differentiable versions. Fogel et al. (2015) explored recovering the linear orderings of items based on pairwise information, another form of permutation optimization. Bengio et al. (2013) introduced the straight-through estimator for differentiating through discrete projections that we utilize in Section 3.3.

### A.3 Experimental Details

#### A.3.1 Multi-layer Perceptron models

In these experiments we utilized networks with 3 hidden layers of 512 units each. ReLU activations were used between layers and no normalization was performed. Optimization was done with Adam and a learning rate of $1e - 3$.

#### A.3.2 VGG-16 and ResNet models on CIFAR datasets

We utilized the VGG-16 architecture of Simonyan & Zisserman (2015) with the exception that we used LayerNorm normalization in place of BatchNorm. Similarly we used the ResNet20 architecture of He et al. (2016) but with LayerNorms in place of BatchNorms.

The following data augmentation was performed during training

- Random resizes of the image between $0.8\times$-$1.2\times$
- Random $32\times32$ pixel crops
- Random horizontal flips
- Random rotations between $\pm30°$

Optimization was done with SGD with momentum (momentum set to 0.9). A weight decay regularization term of $5e - 4$ was applied. A single cosine decay schedule with linear warm-up was used. Learning rates were initialized at $1e - 6$ and linearly increased to $1e - 1$ over the span of an epoch. After that point a single cosine decay schedule (Loshchilov & Hutter, 2017) was used for the remainder of training.

#### A.3.3 ResNet50 models on ImageNet-1K

In this experiment we utilized pre-trained ResNet50 model available for download online, and one trained ourselves. These were standard ResNet50 models, including the use of BatchNorm. In line with prior work (Izmailov et al., 2018; Wortsman et al., 2021; Maddox et al., 2019; Wang et al., 2021), we recalculate BatchNorm statistics after performing weight interpolation. After our initial publication, the recalculation of BatchNorm statistics was suggested to us by the authors of Jordan et al. (2022).

### A.4 The Relationship between Permutation Matching and Normalization Layers

In this section we discuss the impact that different types of common normalization layers can have on the feasibility of model merging.

- **BatchNorm** (Ioffe & Szegedy, 2015) generally breaks after interpolating between weights due to the so-called "variance collapse" problem (Jordan et al., 2022). Therefore, we recommend the recalculation of batch statistics after merging models (Izmailov et al., 2018; Wortsman et al., 2021; Maddox et al., 2019; Wang et al., 2021).
- **LayerNorm** (Ba et al., 2016) is invariant to permutations of units and we found that architectures with LayerNorm can be merged without issue.
- **InstanceNorm** (Ulyanov et al., 2016) also places no restrictions on unit order, and in principle does not present any issues, although we have not run any experiments with it.
- **GroupNorm** (Wu & He, 2020) relies on unit indexes to organize units into groups, and therefore is not invariant to permutations of units. In principle, permutation alignment methods would not work on architectures with GroupNorm, though we have not tested this.

### A.5 Additional Information on Algorithm 1

On currently available hardware (p3.2xlarge AWS instance with an NVIDIA V100 GPU), we observed the following timing results with Algorithm 1,

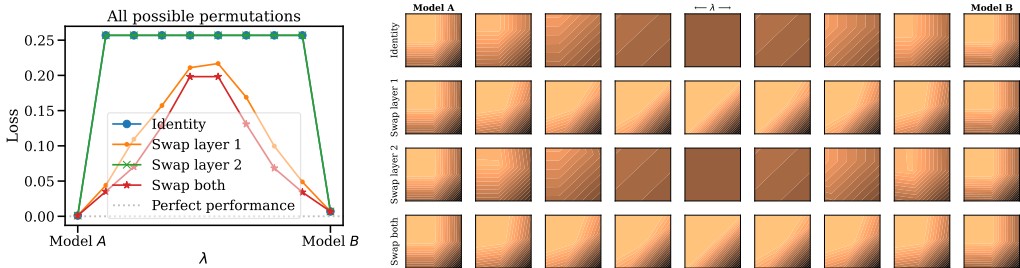

Figure 6: **A counterexample to universal LMC.** There exist models such that no possible permutation of weights allows for linear mode connectivity. *Left:* performance of all possible linear interpolations between the two models. *Right:* A visualization of the prediction functions $f(\boldsymbol{x})$ through each linear sweep. Each row corresponds to one of the four possible permutations, and each column corresponds to a value of $\lambda$, the linear interpolant. The existence of such cases suggests that linear mode connectivity is an artifact of SGD.

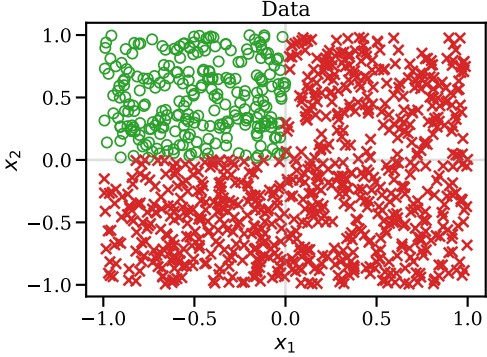

Figure 7: The counterexample classification problem data.

1. MLP (3 layers, 512 units each): 3 seconds

2. ResNet50 ($1\times$ width): 33 seconds

3. ResNet20 ($32\times$ width): 194 seconds

In addition, we tested the ability of Algorithm 1 to recover a known, randomly selected permutation. In a handful of experiments we found that Algorithm 1 was able to exactly recover the known, random permutation in just 3-4 of passes over the layers.

### A.6 COUNTEREXAMPLE DETAILS

Consider a simple 2-dimensional classification task. Our data points are drawn $\boldsymbol{x} \sim$ Uniform$([-1, 1]^2)$ and $y = \mathbf{1}_{x_1 < 0 \text{ and } x_2 > 0}$. Figure 7 provides a visualization of a sample of such data.

We utilize an MLP architecture consisting of two hidden layers, with two units each, and ReLU nonlinearities. Consider two weight assignments that both achieve a perfect fit to the data:

$$f_A(\boldsymbol{x}) = [-1 \quad -1]\, \sigma \left( \begin{bmatrix} -1 & 0 \\ 0 & 1 \end{bmatrix} \sigma \left( \begin{bmatrix} -1 & 0 \\ 0 & -1 \end{bmatrix} \boldsymbol{x} + \begin{bmatrix} 1 \\ 0 \end{bmatrix} \right) + \begin{bmatrix} 1 \\ 0 \end{bmatrix} \right) \tag{4}$$

$$f_B(\boldsymbol{x}) = [-1 \quad -1]\, \sigma \left( \begin{bmatrix} 1 & 0 \\ 0 & -1 \end{bmatrix} \sigma \left( \begin{bmatrix} 1 & 0 \\ 0 & 1 \end{bmatrix} \boldsymbol{x} + \begin{bmatrix} 0 \\ 1 \end{bmatrix} \right) + \begin{bmatrix} 0 \\ 1 \end{bmatrix} \right). \tag{5}$$

We predict a positive label when $f(\boldsymbol{x}) \geq 0$ and a negative label otherwise.

Intuitively, these networks are organized such that each layer makes a classification whether $x_1 < 0$ or $x_2 > 0$. In model $A$, the first layer tests whether $x_2 > 0$, and the second layer tests whether $x_1 < 0$, whereas in model $B$ the order is reversed. With a bit of algebra, it is possible to see that both $f_A$ and $f_B$ achieve perfect performance. However, no possible permutation of units results in linear mode connectivity between $f_A$ and $f_B$. We visualize all possible permutations in Figure 6.

We claim that this example, and the underlying trick, are simple enough to be embedded into larger models. For example, this could trivially be extended to ResNets where different subsets of layers could be set to identity functions.

As discussed in Section 4, the existence of adversarial basins in the loss landscape has interesting consequences for our understanding of loss landscape geometry. In particular, we argue that this implies that common optimization algorithms are conveniently biased towards solutions that admit LMC. However, the precise connection between optimization algorithms and linear mode connectivity remains unclear.

This counterexample does not constitute a contradiction of the conjecture in Entezari et al. (2021). To be more precise, Conjecture 1 of Entezari et al. (2021) proposes that there exists some subset, $\mathcal{S}$, of parameter space such that every pair of elements in $\mathcal{S}$ can be linearly mode connected (after some permutation of units), and that with high probability SGD solutions are contained in $\mathcal{S}$. The example presented in this section does not contradict Entezari et al. (2021)'s conjecture, but instead illustrates that the restriction to SGD solutions is a "load-bearing" element of the conjecture.

### A.7 ON THE FAILURES OF GREEDY UNI-DIRECTIONAL MATCHING

In contrast to prior work (Pittorino et al., 2022; Singh & Jaggi, 2020; Wang et al., 2020), we eschew greedy uni-directional, single-pass matching between models. Instead we derive a weight matching algorithm from a principled, yet computationally infeasible optimization problem. In contrast to prior work, our method can be viewed as "bi-directional": it selects unit associations based on weights in all relevant layers, not just in the immediately previous layer. In this section, we describe benefits of our holistic approach, including an example problem showing the failure modes of greedy uni-directional matching. We find that matching across all layers simultaneously allows our weight matching algorithm (Algorithm 1) to exploit units' relationships with downstream weights in a way that greedy uni-directional matching cannot.

Concretely, greedy uni-directional matching begins at the first layer and computes a matching, $P_1$, considering only $W_1^{(A)}, W_1^{(B)}$. After matching, $P_1$ is applied throughout the remainder of the network. Then, we proceed through the layers in order repeating this process.

We compare our Algorithm 1 to greedy uni-directional matching experimentally in Appendix A.7.1 and present a theoretical counterexample that illustrates the advantages of our method in Appendix A.7.2.

#### A.7.1 EXPERIMENTAL COMPARISON

In order to further evaluate our performance relative to prior work, and Pittorino et al. (2022); Singh & Jaggi (2020) in particular, we explored experimental comparisons with VGG11 models trained on CIFAR-10 and on ResNet50 models trained on ImageNet.

**VGG11 models trained on CIFAR-10.** Following an exact reproduction of experiment from Table 1 of Singh & Jaggi (2020), we merged the model weights released along with their paper. We present results in Table 2. We found that our weight matching method outperforms the "wts" method of Singh & Jaggi (2020) in both implementation speed and model performance when reproducing their experiment with their trained model weights.

**ResNet50 models trained on ImageNet.** We applied OT-Fusion ("wts") to the ImageNet experiment that we consider in Section 5.1. We present results in Figure 8. We found the OT-Fusion method resulted in models with 1.38% top-1 accuracy on ImageNet, only marginally improving over naïve averaging. On the other hand, we achieve 51.01%.

BatchNorm statistics were recalculated after interpolation for all methods shown.

| Method | Test accuracy (↑) | Run-time (↓) |
|---|---|---|
| OT-Fusion (Singh & Jaggi, 2020) | 85.98% | 2.86s |
| Weight matching (ours) | **86.57%** | **0.64s** |

Table 2: **VGG11/CIFAR-10 performance relative to Singh & Jaggi (2020).** We found that our weight matching method outperforms the "wts" method of Singh & Jaggi (2020) in both implementation speed and model performance when reproducing one of their experiments with their published model weights. Our implementation is 4.5× faster, and produces a solution with better model performance.

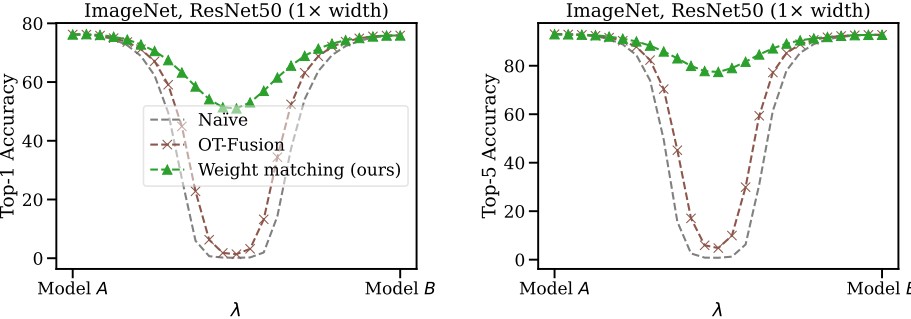

Figure 8: **ResNet50/ImageNet performance relative to Singh & Jaggi (2020).** We found the OT-Fusion method resulted in models with 1.38% top-1 accuracy on ImageNet, only marginally improving over naïve averaging. On the other hand, we achieve 51.01%.

Although a number of factors may be responsible for the difference in performance between weight matching and OT-Fusion, we found the number of alignment passes made over the network layers to have a substantial impact. OT-Fusion is inherently limited to a single pass over the layers. On the other hand, we are not limited to any specific number of optimization passes and instead continue until convergence (convergence is guaranteed by Lemma 2). For comparison, if our weight matching algorithm is artificially handicapped to a single pass over the layers, we achieve a similarly low $\sim 7\%$ top-1 accuracy.

### A.7.2 An Example Failure Case

Consider two networks, $A$ and $B$, with the objective that they capture the identity function $f(x) = x$,

$$f_{\Theta_A}(x) = \begin{bmatrix} 1 & 0 \end{bmatrix} \begin{bmatrix} 1 & 0 \\ 0 & \epsilon \end{bmatrix} \begin{bmatrix} 1 \\ 1 + \epsilon \end{bmatrix} x \tag{6}$$

$$f_{\Theta_B}(x) = \begin{bmatrix} 0 & 1 \end{bmatrix} \begin{bmatrix} 0 & 0 \\ 0 & 1 \end{bmatrix} \begin{bmatrix} 1 \\ 1 + \epsilon \end{bmatrix} x \tag{7}$$

where $\epsilon > 0$ is some negligible constant. It can be seen that these reduce to $f_{\Theta_A}(x) = x$ and $f_{\Theta_B}(x) = (1 + \epsilon)x$.

When aligning these models there are two possible opportunities for permutation, $\pi = \{P_1, P_2\}$. The permuted model $B$ then has the form

$$f_{\pi(\Theta_B)}(x) = \left( \begin{bmatrix} 0 & 1 \end{bmatrix} P_2^\top \right) \left( P_2 \begin{bmatrix} 0 & 0 \\ 0 & 1 \end{bmatrix} P_1^\top \right) \left( P_1 \begin{bmatrix} 1 \\ 1 + \epsilon \end{bmatrix} \right) x \tag{8}$$

Now, aligning with greedy uni-directional matching will result in the alignment $\pi_{gud} = \{P_1 = I, P_2 = I\}$. On the other hand, our weight matching method (Algorithm 1) results in $\pi_{wm} = \left\{ P_1 = \begin{bmatrix} 0 & 1 \\ 1 & 0 \end{bmatrix}, P_2 = \begin{bmatrix} 0 & 1 \\ 1 & 0 \end{bmatrix} \right\}$, regardless of the algorithm's execution order.

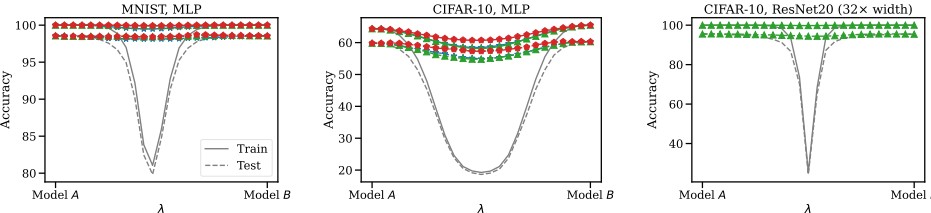

Figure 9: Top-1 accuracy results for the MNIST and CIFAR-10 models of Figure 2.

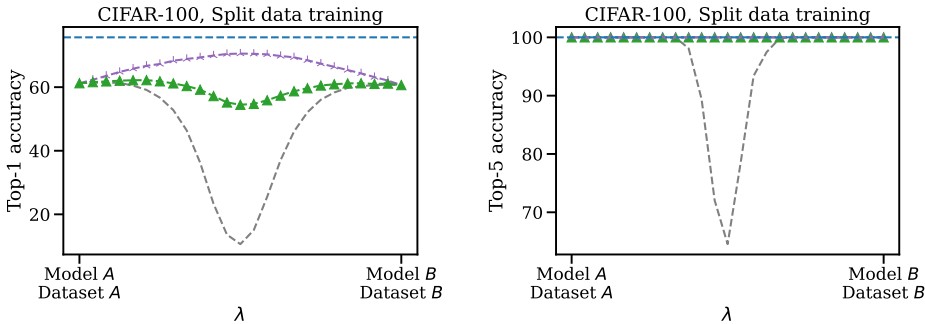

Figure 10: Accuracy results for the CIFAR-100 split data experiment.

Interpolating these matched models at $\lambda = 0.5$, we have

$$f_{\frac{1}{2}(\Theta_A + \pi_{gud}(\Theta_B))}(x) = \begin{bmatrix} 0.5 & 0.5 \end{bmatrix} \begin{bmatrix} 0.5 & 0 \\ 0 & (1+\epsilon)/2 \end{bmatrix} \begin{bmatrix} 1 \\ 1+\epsilon \end{bmatrix} x \qquad = (0.5 + O(\epsilon))x \quad (9)$$

$$f_{\frac{1}{2}(\Theta_A + \pi_{wm}(\Theta_B))}(x) = \begin{bmatrix} 1 & 0 \end{bmatrix} \begin{bmatrix} 1 & 0 \\ 0 & \epsilon/2 \end{bmatrix} \begin{bmatrix} 1+\epsilon/2 \\ 1+\epsilon/2 \end{bmatrix} x \qquad = (1+\epsilon/2)x \quad (10)$$

Here we can see that the greedy uni-directional matching (Equation (9)) results in a merged model that fails to represent the input, identity-function models. On the other hand, our weight matching algorithm (Algorithm 1, Equation (10)) produces a merged model that accurately reflects both of the input models, and even improves performance over the $B$ model. [5]

## A.8 AUXILIARY PLOTS

## A.9 STRAIGHT-THROUGH ESTIMATOR DETAILS

See Algorithm 2 for a complete description of the straight-through estimator algorithm.

## A.10 MERGING MANY MODELS

We propose Algorithm 3 to merge the weights of more than two models at a time.

Following an argument similar to Lemma 2, it can be seen that Algorithm 3 terminates.

In our limited testing, we found that this algorithm converges quickly to solutions that extrapolate better than individual models and results in a merged model with better probability estimate calibra-

---

[5]We note that this example can be extended to the more conventional presentation, including non-linear activation functions, by inserting ReLU activations in each layer and considering $x$ in the positive domain $\mathbb{R}^+$. The positive domain restriction can also be lifted by instead considering the task of absolute value estimation and adding layers $\sigma \circ \begin{bmatrix} 1 & 1 \end{bmatrix} \circ \sigma \circ \begin{bmatrix} 1 \\ -1 \end{bmatrix}$ to the beginning of the network.

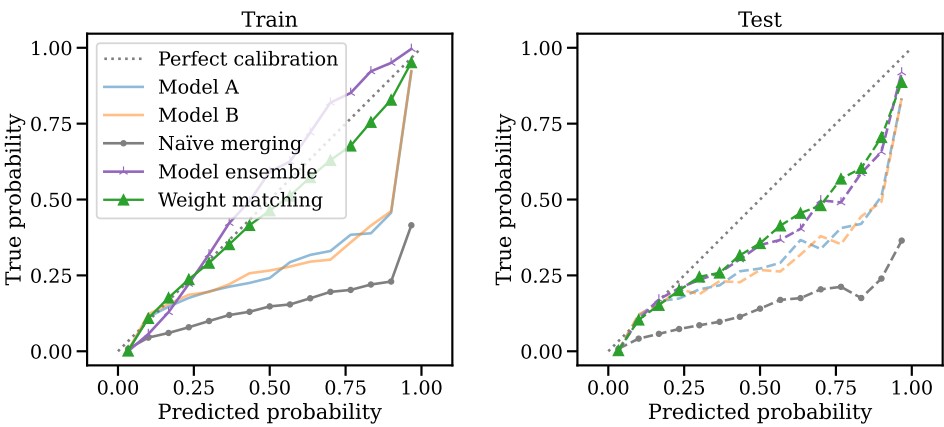

Figure 11: **Merging CIFAR-100 split data models results in superior probability calibration.** Although our merged model is not competitive in terms of top-1 accuracy in the CIFAR-100 split data experiment, we find that it has far better calibrated probability estimates than either of the input models. In addition, we achieve calibration results on par with model ensembling while requiring $2\times$ less memory and compute.

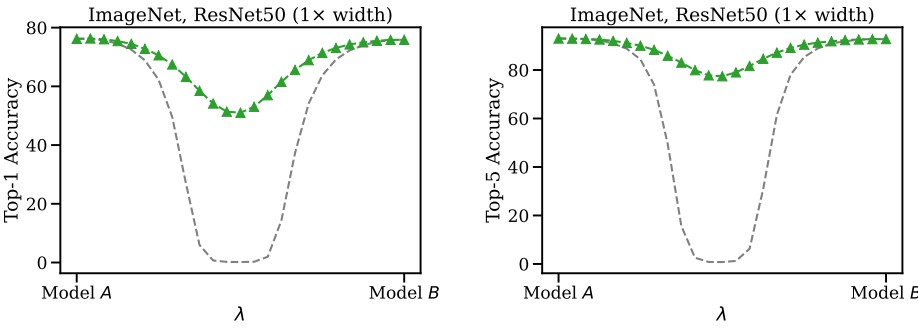

Figure 12: Accuracy results for merged ResNet50 ($1\times$ width) models on ImageNet.

---

**Algorithm 2:** Straight-through estimator training

---

**Given:** Model weights $\Theta_A$, $\Theta_B$, and a learning rate $\eta$.
**Result:** A permutation $\pi$ of $\Theta_B$ such that $\mathcal{L}(\frac{1}{2}(\Theta_A + \pi(\Theta_B)))$ is approximately minimized.

---

**Initialize:** $\tilde{\Theta}_B \leftarrow \Theta_A$
**repeat**
  $\pi(\Theta_B) \leftarrow \text{proj}(\tilde{\Theta}_B)$ using Algorithm 1.
  Evaluate the loss of the midpoint, $\mathcal{L}(\frac{1}{2}(\Theta_A + \pi(\Theta_B)))$.
  Evaluate the gradient, $\nabla\mathcal{L}$, using $\tilde{\Theta}_B$ in place of $\pi(\Theta_B)$ in the backwards pass.
  Update parameters, $\tilde{\Theta}_B \leftarrow \tilde{\Theta}_B - \eta\nabla\mathcal{L}$.
**until** *convergence*

---

---

**Algorithm 3:** MERGEMANY

---

**Given:** Model weights $\Theta_1, \ldots, \Theta_N$

**Result:** A merged set of parameters $\tilde{\Theta}$.

---

**repeat**
    **for** $i \in$ RANDOMPERMUTATION$(1, \ldots, N)$ **do**
        $\Theta' \leftarrow \frac{1}{N-1} \sum_{j \in \{1,\ldots,N\} \setminus \{i\}} \Theta_j$
        $\pi \leftarrow$ PERMUTATIONCOORDINATEDESCENT$(\Theta', \Theta_i)$
        $\Theta_i \leftarrow \pi(\Theta_i)$
    **end**
**until** *convergence*
**return** $\frac{1}{N} \sum_{j=1}^{N} \Theta_j$

---

|  | Train Loss | Train Acc. | Test Loss | Test Acc. |
|---|---|---|---|---|
| Seed 1 | 0.0000 | 1.0000 | 0.1153 | 0.9856 |
| Seed 2 | 0.0000 | 1.0000 | 0.1531 | 0.9854 |
| Seed 3 | 0.0000 | 1.0000 | 0.1229 | 0.9855 |
| Seed 4 | 0.0000 | 1.0000 | 0.1108 | 0.9865 |
| Seed 5 | 0.0000 | 1.0000 | 0.1443 | 0.9871 |
| MERGEMANY | 0.0141 | 0.9952 | **0.0727** | 0.9831 |

Table 3: **Merging multiple models decreases test loss by 43%.** We train five separate MLPs on MNIST. Using Algorithm 3 we merge all these models together simultaneously. This produces a model that appears to have better out-of-distribution performance than any of the input models, with superior test loss performance. We are excited by potential applications of this methodology in federated learning and ensembling, esp. along the lines of "model soups" (Wortsman et al., 2022a).

tion than any of the input models. For example, we present the results of this algorithm on MLPs trained on MNIST in Table A.10.

In addition, we found that merging multiple models helps to calibrate the resulting model predictions. We present this effect in Figure 13.

### A.11 FAILED IDEA: A METHOD FOR STEEPEST DESCENT

Imagine standing in weight space at $\Theta_A$ and trying to decide in which immediate direction to move in order to approach a $\Theta_B$-equivalent point. There are many, many possible permutations of $\Theta_B$ – call them $\pi^{(1)}(\Theta_B), \pi^{(2)}(\Theta_B), \ldots$ – to aim for in the distance. Assuming that the loss landscape is in fact convex modulo these permutation symmetries, a natural choice would be to pick the $\pi^{(i)}(\Theta_B)$ that corresponds to the direction of steepest descent starting from $\Theta_A$ since we expect $\pi^{(i)}(\Theta_B)$ to lie in the same basin as $\Theta_A$. In other words,

$$\min_{\pi} \frac{d \mathcal{L}(\Theta_A + \lambda(\pi(\Theta_B) - \Theta_A))}{d \lambda}\bigg|_{\lambda=0} = \min_{\pi} \nabla \mathcal{L}(\Theta_A)^\top (\pi(\Theta_B) - \Theta_A) \tag{11}$$

$$= -\nabla \mathcal{L}(\Theta_A)^\top \Theta_A + \min_{\pi} \nabla \mathcal{L}(\Theta_A)^\top \pi(\Theta_B) \tag{12}$$

Now, we are tenuously in a favorable situation: $\nabla \mathcal{L}(\Theta_A)$ is straightforward to compute, and picking the best $\pi$ reduces to a matching problem. In particular it is a SOBLAP matching problem of the same form as in Section 3.2. In addition, there is a fast, exact solution for the single intermediate layer case ($L = 2$).

In practice, we found that this method can certainly find directions of steepest descent, but that they are accompanied by high barriers in between the initial dip and $\pi(\Theta_B)$.

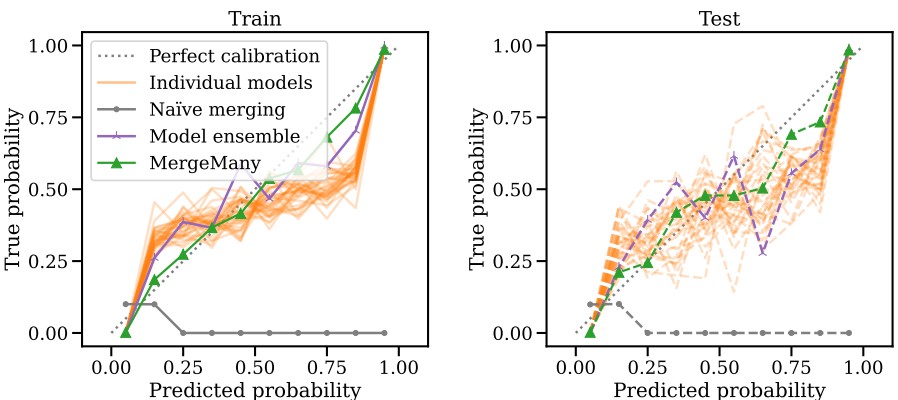

Figure 13: **Merging multiple models results in superior calibration.** Here we show the results of running Algorithm 3 on 32 MLP models trained on MNIST, with each model given access to a random 50% of the training dataset. The resulting merged model demonstrates substantively improved calibration of probability estimates on both the training and test datasets. MergeMany calibration results are competitive with model ensembling, despite requiring $32\times$ less memory and compute.

## A.12 PROOF OF LEMMA 1

To lighten notation we use $\langle \cdot, \cdot \rangle = \langle \cdot, \cdot \rangle_F$ in this section.

**Lemma.** *Given $\boldsymbol{A}, \boldsymbol{B} \in \mathbb{R}^{d \times d}$,*

$$\min_{\boldsymbol{P}, \boldsymbol{Q} \text{ perm. matrices}} \langle \boldsymbol{P} \boldsymbol{A} \boldsymbol{Q}^\top, \boldsymbol{B} \rangle$$

*is strongly NP-hard and has no PTAS.*

*Proof.* We proceed by reduction from the quadratic assignment problem (QAP) (Koopmans & Beckmann, 1957; Cela, 2013). Consider a QAP,

$$\min_{\boldsymbol{P} \text{ perm. matrix}} \langle \boldsymbol{P} \boldsymbol{C} \boldsymbol{P}^\top, \boldsymbol{D} \rangle$$

for $\boldsymbol{C}, \boldsymbol{D} \in \mathbb{R}^{d \times d}$.

Now, pick $\boldsymbol{A} = \boldsymbol{C} + \lambda \boldsymbol{I}$, $\boldsymbol{B} = \boldsymbol{D} - \lambda \boldsymbol{I}$. The we have,

$$\min_{\boldsymbol{P}, \boldsymbol{Q}} \langle \boldsymbol{P}(\boldsymbol{C} + \lambda \boldsymbol{I}) \boldsymbol{Q}^\top, \boldsymbol{D} - \lambda \boldsymbol{I} \rangle = \langle \boldsymbol{P} \boldsymbol{C} \boldsymbol{Q}^\top + \lambda \boldsymbol{P} \boldsymbol{Q}^\top, \boldsymbol{D} - \lambda \boldsymbol{I} \rangle \tag{13}$$

$$= \langle \boldsymbol{P} \boldsymbol{C} \boldsymbol{Q}^\top, \boldsymbol{D} \rangle - \lambda \langle \boldsymbol{P} \boldsymbol{C} \boldsymbol{Q}^\top, \boldsymbol{I} \rangle + \lambda \langle \boldsymbol{P} \boldsymbol{Q}^\top, \boldsymbol{D} \rangle - \lambda^2 \langle \boldsymbol{P} \boldsymbol{Q}^\top, \boldsymbol{I} \rangle \tag{14}$$

$$= \langle \boldsymbol{P} \boldsymbol{C} \boldsymbol{Q}^\top, \boldsymbol{D} \rangle - \lambda \langle \boldsymbol{P}^\top \boldsymbol{Q}, \boldsymbol{C} \rangle + \lambda \langle \boldsymbol{P} \boldsymbol{Q}^\top, \boldsymbol{D} \rangle - \lambda^2 \operatorname{tr}(\boldsymbol{P} \boldsymbol{Q}^\top) \tag{15}$$

For sufficiently large $\lambda$, the $\operatorname{tr}(\boldsymbol{P} \boldsymbol{Q}^\top)$ term will dominate. Letting $\alpha = \max(\max_{i,j} |C_{i,j}|, \max_{i,j} |D_{i,j}|)$, we can bound the other terms,

$$-d^2 \alpha^2 \leq \quad \langle \boldsymbol{P} \boldsymbol{C} \boldsymbol{Q}^\top, \boldsymbol{D} \rangle \leq d^2 \alpha^2 \tag{16}$$

$$-\lambda d \alpha \leq -\lambda \langle \boldsymbol{P}^\top \boldsymbol{Q}, \boldsymbol{C} \rangle \quad \leq \lambda d \alpha \tag{17}$$

$$-\lambda d \alpha \leq \quad \lambda \langle \boldsymbol{P} \boldsymbol{Q}^\top, \boldsymbol{D} \rangle \quad \leq \lambda d \alpha \tag{18}$$

Now there are two classes of solutions: those where $\boldsymbol{P} = \boldsymbol{Q}$ and those where $\boldsymbol{P} \neq \boldsymbol{Q}$. We seek to make the best (lowest) possible $\boldsymbol{P} \neq \boldsymbol{Q}$ solution to have worse (higher) objective value than the worst (highest) $\boldsymbol{P} = \boldsymbol{Q}$ solution. When $\boldsymbol{P} = \boldsymbol{Q}$, the highest possible objective value is

$$d^2 \alpha^2 + \lambda d \alpha + \lambda d \alpha - \lambda^2 d$$

and similarly, the lowest possible objective value when $P \neq Q$ is

$$-d^2\alpha^2 - \lambda d\alpha - \lambda d\alpha - \lambda^2 d + \lambda^2$$

where the final term is due to the fact that at least one entry of $PQ^\top$ must be 0. With some algebra, it can be seen that $\lambda > 5d\alpha$ is sufficient to guarantee that all $P = Q$ solutions are superior to all $P \neq Q$ solutions.

Now when $P = Q$, all frivolous terms reduce to constants and we are left with the QAP objective:

$$\min_P \langle PCP^\top, D \rangle - \lambda \langle P^\top P, C \rangle + \lambda \langle PP^\top, D \rangle - \lambda^2 \operatorname{tr}(PP^\top)$$

$$= -\lambda \operatorname{tr}(C) + \lambda \operatorname{tr}(D) - \lambda^2 d + \min_P \langle PCP^\top, D \rangle$$

completing the reduction. QAP is known to be strongly NP-hard (Koopmans & Beckmann, 1957; Sahni & Gonzalez, 1976) and MaxQAP is known to not admit any PTAS (Makarychev et al., 2014), thus completing the proof. □

### A.13   PROOF OF LEMMA 2

**Lemma.** *Algorithm 1 terminates.*

*Proof.* We proceed by contradiction.

Consider a graph with each possible permutation $\pi_i = \{P_1, \dots, P_{L-1}\}$ as a vertex and directed edges $\pi_i \to \pi_j$ if $\pi_j$ can be reached from $\pi_i$ with a single $P_\ell$ update, as in Algorithm 1. (Ignore those updates that result in no change to $P_\ell$ in order to avoid $\pi_i \to \pi_i$ cycles.) Let $\rho(\pi) = \operatorname{vec}(\Theta_A) \cdot \operatorname{vec}(\pi(\Theta_B))$ denote the utility of a particular $\pi$. Note that $\pi_i \to \pi_j$ implies $\rho(\pi_i) < \rho(\pi_j)$. There exist finitely many possible permutations $\pi_i$, meaning that a failure to terminate must involve a cycle in the graph $\pi_1 \to \cdots \to \pi_n \to \pi_1$. However $\rho$ forms a total order on the vertices and therefore we have a contradiction. □

