# OpenReview forum: "Git Re-Basin: Merging Models modulo Permutation Symmetries"
_ICLR.cc/2023/Conference — ICLR 2023 notable top 5%_

### Official Review · Reviewer_Nc4c · 2022-10-26

**Confidence:** 4
**Correctness:** 4
**Technical Novelty And Significance:** 4
**Empirical Novelty And Significance:** 4
**Recommendation:** 10

**Clarity, Quality, Novelty And Reproducibility:**

**Clarity**

The paper is generally well-written and easy to follow. However, a few details should be clarified:
* Why does Algorithm 1 use a random permutation of layers instead of a uniform curriculum?
* Why does the straight-through estimator consider the loss at the midpoint of the (projected) network parameters instead of a minimax problem over all possible interpolation coefficients $\lambda$?
* Section 4 states that `SGD is implicitly biased towards solutions admitting LMC`. Why is that the case?
* How is the ensembling method in Figure 5 dependent on the interpolation coefficient $\lambda$?
* It is important to emphasize that any method that combines the models in section 5.4 should not rely on data (as in the case of weight matching).

Moreover, the interlude section is difficult to follow since all the results are presented in the Appendix. Therefore, I suggest using the additional page of the camera-ready version to move the result to the main paper.

**Quality**

The proposed methods are sound and efficient. Moreover, the experimental evaluation is exhaustive and showcases the effectiveness of the methods.

**Novelty**

To the best of my knowledge, the paper is the first to propose permutation selection methods to demonstrate linear mode connectivity of neural networks.

**Reproducibility**

The paper provides sufficient details (including code, logs, and model checkpoints) of the experimental setup to reproduce the results.


**Strength And Weaknesses:**

**Strengths**

The paper investigates an essential question in optimizing neural networks: Why do many local minima exist in the loss landscape with similar performance? The paper provides strong evidence for the conjecture that these minima all correspond to the same network modulo permutation symmetries, which provides a significant boost to the community's understanding of gradient descent landscapes. To that end, the paper introduces three novel and computationally efficient permutation selection methods that permute a model's activations or weights to match a target network's behavior. The methods are grounded in concepts from combinatorial optimization and are equipped with convergence guarantees. The paper opens up a variety of exciting research directions:
* Can the methods be extended for models with slightly different architectures?
* Does the method also work for models trained on fundamentally different datasets?
* Can we leverage these methods to combine different networks specializing in specific tasks to create a network capable of performing all tasks simultaneously?
* Can we leverage these methods to interpolate between two models in a subtractive manner, e.g., subtract a biased model from an accurate model without loss in accuracy?

**Weaknesses**

The paper claims that `neural network loss landscapes contain (nearly) a single basin after accounting for all possible permutation symmetries of hidden units`. However, the paper only demonstrates that the proposed permutation selection methods induce linear mode connectivity for two given networks (as far as I can tell from the experiment description, the results have not been aggregated over multiple independent runs). As a result, the paper can only reasonably claim that `for some networks, the permutation symmetries allows projecting them into the same loss basin`. Thus, the claim either has to be reformulated, or the experiments have to be repeated for many network initialization parameters to provide evidence for the claim in its current form.

Moreover, some experimental details could be more precise (see clarity below). Finally, the paper posits a hypothesis on the width of models for ImageNet in section 5.1, which is relatively straightforward to test, but does not conduct a corresponding investigation.

**Summary Of The Paper:**

The paper investigates the linear mode connectivity of neural networks trained with stochastic gradient descent, i.e., the possibility of smoothly interpolating between two (equivalence classes of) networks trained with different initialization parameters (i.e., one network is permuted to lie in the same loss basin as the other). To that end, the paper proposes three novel algorithms that compute a network permutation that matches another network's behavior (e.g., activations or weights). The approaches are based on regression between activations, approximate weight matching, and straight-through estimation. Finally, the paper conducts an empirical investigation of various architectures and datasets to demonstrate that the proposed methods successfully interpolate between two networks without increasing the loss.

**Summary Of The Review:**

Given the significance of the results, the potential impact on the community, and the quality of the paper, I strongly recommend acceptance of the paper.

---

> ### Public Comment · ~Sidak_Pal_Singh1 · 2022-11-06
> **On Git Re-Basin**
>
> Thank you for your review! We’d appreciate it if on the novelty aspect of their ‘three proposed novel algorithms’ you would be so kind as to peruse our general comment https://openreview.net/forum?id=CQsmMYmlP5T&noteId=9liIVMeFFnW on this paper, as well as the concerns mentioned by Reviewer qnqB. Moreover, if you’d allow us to mention, the methods of [2-4] can already handle architectures of different widths with ease (see, section 5.2 of [3, https://arxiv.org/pdf/1910.05653.pdf], with mentioned applications to pruning and federated learning). Since [1-4], Nguyen et al. (2021) https://arxiv.org/pdf/2110.15538.pdf can now support even unequal depth networks.

---

> > ### Author Response · Authors · 2022-11-12
> >
> > See our response [here](https://openreview.net/forum?id=CQsmMYmlP5T&noteId=au5C96yfoHB).

---

> ### Author Response · Authors · 2022-11-12
>
> Thank you for your wonderful review and feedback!
>
> > The paper claims that neural network loss landscapes contain (nearly) a single basin after accounting for all possible permutation symmetries of hidden units. However, the paper only demonstrates that the proposed permutation selection methods induce linear mode connectivity for two given networks (as far as I can tell from the experiment description, the results have not been aggregated over multiple independent runs). As a result, the paper can only reasonably claim that for some networks, the permutation symmetries allows projecting them into the same loss basin. Thus, the claim either has to be reformulated, or the experiments have to be repeated for many network initialization parameters to provide evidence for the claim in its current form.
>
> We have found that our results tend to be robust across repeated runs. We simply used seeds 0 and 1 when training models A and B in each of our experiments, so no cherry picking has been done there. On the other hand, we have occasionally observed some brittleness depending on peculiarities of the experimental setup. We discuss these cases in Appendix A.1.
>
> > Moreover, some experimental details could be more precise (see clarity below). Finally, the paper posits a hypothesis on the width of models for ImageNet in section 5.1, which is relatively straightforward to test, but does not conduct a corresponding investigation.
>
> Agreed! Sadly, training wider ImageNet models is beyond our compute capabilities at the moment. In fact, we were only able to run the ImageNet experiment thanks to pre-trained weights being available for download online. This could be an exciting experiment for someone with more compute resources!
>
> ### Clarity questions:
> > Why does Algorithm 1 use a random permutation of layers instead of a uniform curriculum?
>
> We selected random permutations of layers so as to not bias the algorithm towards preferring matching to the initial layers over later layers. The order of layer alignments matters since alignment choices made on a layer in one step will bias the alignments of other layers in future steps. We avoided having to pick a “first” layer first by simply letting the order be randomly assigned on every iteration.
>
> > Why does the straight-through estimator consider the loss at the midpoint of the (projected) network parameters instead of a minimax problem over all possible interpolation coefficients λ?
>
> Good question! A minimax phrasing of the optimization would be more thorough but also more computationally expensive. Selecting λ = ½ is intentional: Thanks to A/B symmetry between the models, the loss interpolation curve will be symmetric across the x-axis in expectation. Since the endpoints (λ = 0, 1) have zero barrier, we can expect the maximum barrier to lie in the middle, at λ = ½.
>
> > Section 4 states that SGD is implicitly biased towards solutions admitting LMC. Why is that the case?
>
> In Sec. 4 we demonstrate a counterexample such that no possible permutation results in LMC. This counterexample is small enough that it can be “embedded” into nearly any larger architecture. We conclude therefore that SGD (and related optimization methods) conveniently tend not to land in these adversarial basins.
>
> However, we hear your feedback regarding this section, and will attempt to improve its clarity!
>
> > How is the ensembling method in Figure 5 dependent on the interpolation coefficient λ?
>
> We ensemble by linearly interpolating between the A and B logits, where λ is the linear interpolation coefficient.
>
> > It is important to emphasize that any method that combines the models in section 5.4 should not rely on data (as in the case of weight matching).
>
> Great point! We have edited the paper to emphasize this requirement.
>
> Thanks again for your review!

---

### Official Review · Reviewer_qnqB · 2022-10-26

**Confidence:** 5
**Correctness:** 3
**Technical Novelty And Significance:** 3
**Empirical Novelty And Significance:** 3
**Recommendation:** 8

**Clarity, Quality, Novelty And Reproducibility:**

## Novelty and Related Work (W1)

Here I will describe some of the relevant methods, results and observations in prior work.

- The fact that the neural network optima found by SGD form a single connected basin has been well established in the literature, to the best of my knowledge first demonstrated by [1, 2] (cited by the paper). The observation has been extended in many ways, e.g. to multi-dimensional mode-connecting constructions by [5] and with a formal proof of mode connectivity [4]. In particular, [1] shows that modes without permutations can be connected by a polygonal chain with a single bend (two line segments). Relative to this observation, the present work shows that if we select an optimal permutation, we can connect the modes with one line segment instead of two.

- In Appendix A.7 of paper [1], the authors show that as the width of the model becomes larger, the mode connecting paths (without permutation) become closer to a line segment, and mode connectivity improves. This experiment is relevant to Section 5.3.

- The work [6] actually finds the optimal weight permutations for optimizing the mode connectivity, which is very closely related to Git-Rebasin. The one difference is that the authors of [6] do not claim linear mode connectivity. However, linear mode connectivity doesn't always hold, so the distinction between the two papers is not as obvious. In particular, both papers develop similar algorithms for finding optimal weight perturbations. I encourage the authors to discuss the differences with [6] in more detail.

- Merging models with weight perturbations has also been considered before in [7] with similar experiments to Figure 5.

Many of these works are cited in the paper, but very briefly. To me, Git-Rebasin combines many of these prior observations (and adds new ones!) into a single unifying picture which is highly valuable. At the same time, the novelty is not _as high_ as it may initially appear.

Other works that could be cited or discussed more:
- [4] shows that weight averaging can be used to merge close-by models and improve generalization. [8] actually more or less contradicts the Git-Rebasin hypothesis, as they argue that modes that are linearly connected also are similar functionally, and functionally distinct models cannot be linearly connected. [9] discusses another symmetry in neural network parameterization relevant to loss surface analysis.


## Experiments (W2)

While the main results are exciting, it appears that there are limitations to the linear mode connectivity observation, discussed in A.1. In particular, it seems like it is fairly easy to construct practically relevant neural network optima pairs for which linear mode connectivity will not hold after Git-Rebasin. The standard mode connectivity results of [1, 2] appear to be more robust.

Similarly, while Figure 5 shows improved test loss, compared to the models being merged, the test accuracy is not improved (Figure 9), and is significantly lower than the combined data accuracy. The improvement in the loss is not necessarily practically relevant, as it probably comes from reducing the confidence in predictions (leading to better calibration), which can be achieved in other ways (e.g. logit tempering [10]).


## Interpretations (W3)

The authors state many times that the linear mode connectivity after Git-Rebasin is connected to the inductive biases of SGD. However, as far as I understand, there are no experiments that test this intepretation. The results is presented based on the observations that (1) there can be models where linear mode connectivity would not hold as shown in Section 4 and (2) linear mode connectivity improves during training. (1) appears to be somewhat irrelevant to SGD, as it is not shown that SGD would not find the solutions described in Section 4. (2) is not directly testing SGD either, and is confounded by the loss value; an alternative explanation is that the linear mode connectivity is more likely to hold for pairs of solutions with low loss.

If the authors want to push for the SGD inductive bias interpretation, I think they should consider experiments that e.g. try to apply Git-Rebasin to models found by non-SGD-like optimizers, e.g. full-batch training, or even L-BFGS.

**Strength And Weaknesses:**

## Strengths

This is a genuinely exciting paper. I believe it will inspire further research into understanding the loss landscapes as well as merging models and related areas. I think the paper can potentially be quite impactful.

As I will argue below, I think many of the pieces of this work have been explored before. To me, the main strength of this paper is that it provides a new perspective, or way of thinking about the loss landscapes, even if somewhat obvious in retrospect.

## Weaknesses

I like this paper, so the weaknesses that I point out are somewhat nit-picky.

**W1**: Many of the key pieces of this work have been considered before separately. I think some additional discussion of related work is needed.

**W2**: It appears that the linear mode connectivity results may be somewhat brittle.

**W3**: Some of the interpretations about the SGD inductive biases are not very well supported by the experiments in my opinion.


**Summary Of The Paper:**

The paper proposes 3 methods for permuting the weights of a neural network $A$ to closely match a second neural network $B$. With this procedure, they show that often they achieve linear mode connectivity between the permuted model $A$ and model $B$. They further show that it is possible to merge independently trained models $A, B$ by averaging the weights of the permuted model $A$ and model $B$.

**Summary Of The Review:**

Overall, really exciting paper. I pointed out some relatively minor weaknesses, so I currently vote for 8.

## References

[1] [_Loss Surfaces, Mode Connectivity, and Fast Ensembling of DNNs_](https://arxiv.org/abs/1802.10026);
Timur Garipov, Pavel Izmailov, Dmitrii Podoprikhin, Dmitry Vetrov, Andrew Gordon Wilson

[2] [_Essentially No Barriers in Neural Network Energy Landscape_](https://arxiv.org/abs/1803.00885);
Felix Draxler, Kambis Veschgini, Manfred Salmhofer, Fred A. Hamprecht

[3] [_Explaining Landscape Connectivity of Low-cost Solutions for Multilayer Nets_](https://arxiv.org/abs/1906.06247);
Rohith Kuditipudi, Xiang Wang, Holden Lee, Yi Zhang, Zhiyuan Li, Wei Hu, Sanjeev Arora, Rong Ge

[4] [_Averaging Weights Leads to Wider Optima and Better Generalization_](https://arxiv.org/abs/1803.05407);
Pavel Izmailov, Dmitrii Podoprikhin, Timur Garipov, Dmitry Vetrov, Andrew Gordon Wilson

[5] [_Loss Surface Simplexes for Mode Connecting Volumes and Fast Ensembling_](https://arxiv.org/abs/2102.13042);
Gregory W. Benton, Wesley J. Maddox, Sanae Lotfi, Andrew Gordon Wilson

[6] [_Optimizing Mode Connectivity via Neuron Alignment_](https://arxiv.org/abs/2009.02439);
N. Joseph Tatro, Pin-Yu Chen, Payel Das, Igor Melnyk, Prasanna Sattigeri, Rongjie Lai

[7] [_Model Fusion via Optimal Transport_](https://arxiv.org/abs/1910.05653);
Sidak Pal Singh, Martin Jaggi

[8] [_Linear Connectivity Reveals Generalization Strategies_](https://arxiv.org/abs/2205.12411);
Jeevesh Juneja, Rachit Bansal, Kyunghyun Cho, João Sedoc, Naomi Saphra

[9] [_Sharp Minima Can Generalize For Deep Nets_](https://arxiv.org/abs/1703.04933);
Laurent Dinh, Razvan Pascanu, Samy Bengio, Yoshua Bengio

[10] [_On Calibration of Modern Neural Networks_](https://arxiv.org/abs/1706.04599);
Chuan Guo, Geoff Pleiss, Yu Sun, Kilian Q. Weinberger

---

> ### Public Comment · ~Sidak_Pal_Singh1 · 2022-11-06
> **On Git Re-Basin**
>
> Thank you for your review! We share several of the novelty concerns you mentioned and included more details in our comment here https://openreview.net/forum?id=CQsmMYmlP5T&noteId=9liIVMeFFnW.

---

> > ### Author Response · Authors · 2022-11-12
> >
> > See our response [here](https://openreview.net/forum?id=CQsmMYmlP5T&noteId=au5C96yfoHB).

---

> ### Author Response · Authors · 2022-11-12
>
> Thank you for your detailed feedback and kind words!
>
> # W1: Many of the key pieces of this work have been considered before separately. I think some additional discussion of related work is needed.
>
> We humbly accept your feedback regarding additional discussion of related work. As a result of your feedback, we have greatly expanded our discussion of related work with an Extended Related Work section in Appendix A.2.
>
> > The fact that the neural network optima found by SGD form a single connected basin has been well established in the literature, to the best of my knowledge first demonstrated by [1, 2] (cited by the paper). The observation has been extended in many ways, e.g. to multi-dimensional mode-connecting constructions by [5] and with a formal proof of mode connectivity [4]. In particular, [1] shows that modes without permutations can be connected by a polygonal chain with a single bend (two line segments). Relative to this observation, the present work shows that if we select an optimal permutation, we can connect the modes with one line segment instead of two.
>
> [1, 2, 5] propose viewing trained solutions as being “mode connected” via non-linear paths. However – as you note – we pursue strictly linear paths between solutions after accounting for permutation symmetries.
>
> We agree that [1, 2, 5] are indeed fantastic prior work! However, identifying non-linear mode connecting paths tends to be quite a bit slower than our proposed weight matching method, making applications to efficient model merging more challenging. We hope that our work may contribute additional insights to the study of mode connectivity and loss landscape geometry.
>
> > In Appendix A.7 of paper [1], the authors show that as the width of the model becomes larger, the mode connecting paths (without permutation) become closer to a line segment, and mode connectivity improves. This experiment is relevant to Section 5.3.
>
> Thank you for bringing these results to our attention! We were previously unaware of this particular section, and we have added a note regarding it to our Extended Related Work (Appendix A.2).
>
> > The work [6] actually finds the optimal weight permutations for optimizing the mode connectivity, which is very closely related to Git-Rebasin. The one difference is that the authors of [6] do not claim linear mode connectivity. However, linear mode connectivity doesn't always hold, so the distinction between the two papers is not as obvious. In particular, both papers develop similar algorithms for finding optimal weight perturbations. I encourage the authors to discuss the differences with [6] in more detail.
>
> Thank you for bringing this to our attention! Since discovering that the activation matching method of Sec. 3.1 is similar to prior work [6, 7] we have updated our paper to include a note highlighting this connection. Still, our derivation following a least-squares regression framework is new as far as we are aware.
>
> In addition, it is our understanding that the primary paradigm in [6] is to use matching in service of finding a non-linear mode connecting path. In contrast, we focus strictly on the linear mode connectivity case.
>
> > Merging models with weight perturbations has also been considered before in [7] with similar experiments to Figure 5.
>
> This is true. However, we found our weight matching method to be superior to [7] in terms of both speed and merged model performance. For example, [7]’s method gets 1.38% top-1 validation set accuracy on ImageNet, while our method achieves 51.01%. We discuss the relationship with [7] in more detail in [in this comment](https://openreview.net/forum?id=CQsmMYmlP5T&noteId=au5C96yfoHB) and Appendix A.7, and have included a more thorough discussion of [7] in our Extended Related Work (Appendix A.2) in response to your feedback.
>
> # W2: It appears that the linear mode connectivity results may be somewhat brittle.
>
> > While the main results are exciting, it appears that there are limitations to the linear mode connectivity observation, discussed in A.1. In particular, it seems like it is fairly easy to construct practically relevant neural network optima pairs for which linear mode connectivity will not hold after Git-Rebasin. The standard mode connectivity results of [1, 2] appear to be more robust.
>
> Indeed, we found that linear mode connectivity is substantially more challenging than non-linear mode connectivity as in [1, 2]. Although of course we would have preferred to find perfect linear mode connectivity across the board, we believe that the failure modes of LMC constitute an intriguing phenomenon and studying them could be a potentially fertile avenue for future work.
>
> We also highlight limitations of our work in Sec. 4 and Appendix A.1.

---

> > ### Author Response · Authors · 2022-11-12
> >
> > # W3: Some of the interpretations about the SGD inductive biases are not very well supported by the experiments in my opinion.
> >
> > > The authors state many times that the linear mode connectivity after Git-Rebasin is connected to the inductive biases of SGD. However, as far as I understand, there are no experiments that test this intepretation. The results is presented based on the observations that (1) there can be models where linear mode connectivity would not hold as shown in Section 4 and (2) linear mode connectivity improves during training. (1) appears to be somewhat irrelevant to SGD, as it is not shown that SGD would not find the solutions described in Section 4. (2) is not directly testing SGD either, and is confounded by the loss value; an alternative explanation is that the linear mode connectivity is more likely to hold for pairs of solutions with low loss.
> >
> > If the authors want to push for the SGD inductive bias interpretation, I think they should consider experiments that e.g. try to apply Git-Rebasin to models found by non-SGD-like optimizers, e.g. full-batch training, or even L-BFGS.
> >
> > This is an interesting point! To clarify, our intended argument was that adversarial basins exist that cannot be linearly mode connected under any permutation of weights. We demonstrate this with a counterexample small enough that it can be “embedded” into nearly any larger model architecture. Our conclusion is therefore that deep learning optimization methods (conveniently) do not tend to land in these adversarial basins. Since SGD and its brethren – Adam, SGD with momentum, and so forth – are the predominant methods for deep learning optimization we implicated SGD, while referring to training procedures more holistically.
> >
> > We take your point and have updated the paper to implicate training procedures in place of SGD.
> >
> > Thanks again for your review!

---

> > > ### Comment · Reviewer_qnqB · 2022-12-02
> > > **Thank you for your response**
> > >
> > > Dear authors, thank you for your response and updating the paper. I have read the (expansive) discussion, reviews and responses. I still maintain my assessment.
> > >
> > > I believe that the main contribution of this paper is not algorithmic, but rather conceptual. Indeed, similar methods existed before as pointed out by the discussion and reviews, and if I were to evaluate the paper purely as a model fusion paper, I would probably give it a lower score. However, I believe that this paper provides an interesting new perspective on the structure of the loss surfaces of neural networks, which is its main contribution to me. I agree with the commenters that the paper should acknowledge the factual similarities with prior work where appropriate, but I still believe that the paper brings value to the community and *should be accepted to the conference*.
> > >
> > > I would also like to note that while the discussion had scientific value and ultimately made the paper better, the tone of some of the public comments was overly aggressive.

---

### Official Review · Reviewer_fSNF · 2022-10-27

**Confidence:** 4
**Correctness:** 4
**Technical Novelty And Significance:** 3
**Empirical Novelty And Significance:** 3
**Recommendation:** 8

**Clarity, Quality, Novelty And Reproducibility:**

Clarity: The paper states the conjecture, algorithms and experimental results clearly.
Originality: The results shown in the paper are novel.
Reproducibility: I believe the authors provide sufficient information to reproduce the results in the paper.

**Strength And Weaknesses:**

Strengths:
- This paper studies an important open conjecture from Entezari et. al. and provides substantial evidence for it, while stating caveats clearly (requirement of high width)
- The proposed algorithms to find the permutations are interesting. I am surprised that the weight matching algorithm works at all, since it can speed up the compute required to solve the permutation problem by a large amount.
- The paper is well-written


Weaknesses:
- I find the claims regarding the practical utility of merging models trained in a distributed way slightly exaggerated. You don’t see the same improvements in the combined model’s accuracy (Figure 9), and the combined model does much worse than ensembling. Moreover, even ensembling would be a weak baseline if you are trying to combine many models that have been trained on smaller splits of the data. Thus, it is hard for me to see how this method could be helpful in federated learning or distributed training. Nevertheless, I don’t think this particular claim is central to the paper and the results are interesting anyway.

Neutral questions
- Do the authors have any intuitions for why Figure 3b shows so many bumps? Perhaps the training of one of the models is slower? Maybe the right comparison would be interpolating models of approximately equal loss even if they are slightly shifted epochs.
- What are the two different lines in each panel of Figure 2?



**Summary Of The Paper:**

This paper provides further evidence for the conjecture from Entezari et. al. 2021 that solutions found by SGD are linearly mode connected modulo permutations. They provide three methods to find these permutations, one of which (weight matching) is very fast to compute. Using these methods they show that high-width neural networks achieve low loss barriers for tasks like MNIST and CIFAR-10.

**Summary Of The Review:**

I think this paper studies an important conjecture and clearly provides evidence in support of it. I recommend this paper for acceptance.

---

> ### Public Comment · ~Sidak_Pal_Singh1 · 2022-11-06
> **On Git Re-Basin**
>
> Thank you for your review! We’d appreciate it if on the novelty aspect of their proposed algorithms you would be so kind as to peruse our general comment https://openreview.net/forum?id=CQsmMYmlP5T&noteId=9liIVMeFFnW, as well as the concerns mentioned by Reviewer qnqB. Moreover, given your interest, we would also like to invite you to consider the running time efficiency of the pre-existing — but not discussed — algorithms in point 3 of Section (C) https://openreview.net/forum?id=CQsmMYmlP5T&noteId=T3EcmUY0qcV.

---

> > ### Author Response · Authors · 2022-11-12
> >
> > See our response [here](https://openreview.net/forum?id=CQsmMYmlP5T&noteId=au5C96yfoHB).

---

> ### Author Response · Authors · 2022-11-12
>
> Thank you for your careful review of our work and excellent feedback!
>
> > I find the claims regarding the practical utility of merging models trained in a distributed way slightly exaggerated. You don’t see the same improvements in the combined model’s accuracy (Figure 9), and the combined model does much worse than ensembling. Moreover, even ensembling would be a weak baseline if you are trying to combine many models that have been trained on smaller splits of the data. Thus, it is hard for me to see how this method could be helpful in federated learning or distributed training. Nevertheless, I don’t think this particular claim is central to the paper and the results are interesting anyway.
>
> Yes, federated learning and distributed training are motivating applications for our work. In particular, the current status quo in federated learning is to periodically average model weights across all clients using naïve averaging. Previous federated learning works have explored improvements over naïve averaging (eg., “Federated Learning with Matched Averaging“ by Wang et al 2020). Given the speed of our weight matching technique and its improvements over naïve averaging, the intended application would be for weight matching to be used in place of naïve averaging and run periodically throughout training.
>
> However, we humbly accept your feedback and would be happy to revise any particular statements that you think may be exaggerated.
>
> > Do the authors have any intuitions for why Figure 3b shows so many bumps? Perhaps the training of one of the models is slower? Maybe the right comparison would be interpolating models of approximately equal loss even if they are slightly shifted epochs.
>
> Great question! We found that this sort of behavior is indicative of a model having insufficient capacity, esp. in terms of width. In this particular experiment we applied an underpowered MLP on CIFAR-10, so that we would be able to reasonably store model checkpoints at every epoch within our limited storage. Provided more storage it could be exciting to re-run this experiment with a much larger model.
>
> > What are the two different lines in each panel of Figure 2?
>
> We plot results on both train (solid lines) and test (dashed lines) where computationally feasible.
>
> Thanks again for your review!

---

### Public Comment · ~Sidak_Pal_Singh1 · 2022-11-06
**Git Clone or Git Re-Basin? (1/3)**

Dear ICLR community, authors and reviewers,
\
&nbsp;

It is with great regret that we need to bother you for a case where we believe *misinformation is being seeped into the scientific process*.

> In short, we wish to demonstrate that this work, Git Re-Basin, is but a mere **rehash** *(clone?)*  of past works [1-5].

\
&nbsp;
More concretely, we aim to convey the following:
- Method 1 of Git Re-Basin is **provably identical** to the activations-based alignment of OTFusion [3].
- Methods 2 and 3 of Git Re-basin are **highly similar** to the approaches in past work [1,2,3,4] and *yet not a single comparison* to baselines has been performed.
- Many of the results or observations *have already been shown in similar forms* in past works, **but are presented anew** in this work.
- The paper cites some of these works [3-5] as *mere lip service*, while **not accurately relating key contributions**.
- Lastly, in light of the evidence above, we conclude that numerous *claims in the paper are exaggerated, invalid, or deceptive*.

\
&nbsp;
*We want to stress that there is absolutely nothing wrong with building on related work, but one must be honest and not deceptive.* Presently, their *premier ‘contribution of novel algorithms’ will not be novel*; the second regarding SGD seems to be already *under some contention* as alluded to by Reviewers qnqB and Nc4c;  and the third about empirical results with MLPs/CNNs/ResNets on MNIST, CIFAR10, and CIFAR100 — *were more or less already known* [2-6].

----
\
&nbsp;
### (A) Mathematical proof of equivalence to [3]

What better way to start than by showing that Method 1, based on matching the activations, of Git Re-Basin can be mathematically proved to be identical to one of the approaches used in Singh & Jaggi (2019) [3]. The main idea of [3] is to utilize Optimal Transport (OT) to first obtain a layerwise alignment of the given networks and then, post-alignment, average their parameters respectively.

The gist of the proof below (which is fairly straightforward, and can be found in more generality for other kinds of costs in Corollary 1 https://mathematical-tours.github.io/book-sources/optimal-transport/CourseOT.pdf) is that for uniform marginals and Euclidean ground metric, the solution set of both Method 1 and OT-based approach of [3] is identical owing to the classic Birkhoff-von Neumann Theorem.

To recap, the OT problem can be expressed as the following linear program:
$$\underset{{\bf{T}} \in {\mathbb{R}}\_{+}^{d\times d},{\bf{T}}\bf{1}=\alpha,{\bf{T}}^\top\bf{1}=\beta}{\operatorname{\mathop{\mathrm{argmin}}\limits}}\quad \langle {\bf{T}}, {\bf{C}} \rangle_F$$
where, the transport plan ${\bf{T}} \in {\mathbb{R}\_{+}^{d \times d}}$ indicates how much of the ‘goods’ be moved from a ‘source location' to a ‘destination location', and must satisfy the mass-conservation constraints at the source and the destination, i.e.,  ${\bf{T}}\bf{1}=\alpha$ and ${\bf{T}}^\top\bf{1}=\beta$. Further, the ground cost matrix is denoted as ${\bf{C}}$ and the entries of which specify the per-unit transportation cost between the corresponding source-destination pair.

For simplicity, in [3], all neurons are assumed to be of equal importance (same amount of ‘supplies’ and ‘demands’ in the above parlance), and thus we can set $\alpha=\beta=\frac{1}{d}\bf{1}$. For convenience, let us call the Transport map $\bf{T}$ multiplied by the scalar $d$ as $\bf{P}$, i.e., ${\bf{P}} = d  {\bf{T}}$, and $\bf{P}$ now is a bistochastic matrix (all rows and columns must sum to $1$).

Now, let us consider the activation-based approach of [3], where the cost matrix $C$ can be represented, using the notation of Git Re-Basin, as $c_{ij} =\||{\bf{Z}}^{(A)}\_{i,:} - {\bf{Z}}^{(B)}\_{j,:}\||^2_2$.


Then using the shorthand $\bf{z}\_A  = \operatorname{\mathop{\mathrm{diag}}}\left({{\bf{Z}}^{(A)}} {{\bf{Z}}^{(A)}}^\top\right)$ and $\bf{z}\_B = \operatorname{\mathop{\mathrm{diag}}}\left({{\bf{Z}}^{(B)}} {{\bf{Z}}^{(B)}}^\top\right)$ for extracting the diagonal (vector) of the respective matrices, we can express the cost matrix $C$ alternatively as:
  $$ {\bf{C}}=\bf{z}\_A \bf{1}^\top + \bf{1}{\bf{z}\_B^\top} - 2  {\bf{Z}}^{(A)} {{\bf{Z}}^{(B)}}^\top.$$

So, we can express the OT problem as follows:
  $$  \underset{{\bf{P}} \in {\mathbb{R}}\_{+}^{d\times d},{\bf{P}}\bf{1}=\bf{1}, {\bf{P}}^\top\bf{1}=\bf{1}}{\operatorname{\mathop{\mathrm{argmin}}\limits}}\quad {\langle {\bf{P}}, \bf{z}\_A \bf{1}^\top + \bf{1}{\bf{z}\_B}^\top - 2  {\bf{Z}}^{(A)} {{\bf{Z}}^{(B)}}^\top\rangle}_F.  $$

  Now, using the mass-conservation constraints, we get the following equivalent problem
  $$  \underset{{\bf{P}} \in {\mathbb{R}}\_{+}^{d\times d},{\bf{P}}\bf{1}=\bf{1}, {\bf{P}}^\top\bf{1}=\bf{1}}{\operatorname{\mathop{\mathrm{argmin}}\limits}}\quad - \langle {\bf{P}},  {\bf{Z}}^{(A)} {{\bf{Z}}^{(B)}}^\top\rangle_F.  $$

Hence, we recover that the optimization objective above is identical to that in Eqn1. of this work.

*(continued)*

---

> ### Public Comment · ~Sidak_Pal_Singh1 · 2022-11-06
> **Git Clone or Git Re-Basin? (3/3)**
>
> \
> &nbsp;
>
> 2. Next, the authors discuss that
>    >"Additionally, we **identify intriguing phenomena** relating model width and training time to mode connectivity."
>
>    But, *an entire section in [3] details* (Appendix S10 & Table S11) the point of how width decreases the gap in performance for fully-connected networks (post permutation). This is even before [6] showed that wider models exhibit better LMC for both fully-connected and convolutional networks.
>
>    Moreover, the 'time to mode connectivity' can also be seen in a nascent form from Table S1 of [3]. Since the barrier is defined as max over $\lambda$ in $[0, 1]$, obviously, the barrier should be at least as big as the one for $\lambda=0.5$ as described in Table S1.
>
> 3.  Then the authors (wrongly) claim that they are the first to find the solutions within seconds,
>   >**“Unlike previous works** on LMC we accomplish zero-barrier paths between two independently-trained models with an algorithm that runs **on the order of seconds**”, "Our fastest method identifies permutations in **mere seconds**”
>
>     Given that their ‘novel’ algorithms are but rehashes of past works [1-4], this is not surprising when similarly on **page 5 of [3]** it is mentioned that **“the time taken to fuse six VGG11 models is ≈ 15 seconds on 1 Nvidia V100 GPU”**. *Further, in Section S1.4, [3] details that to fuse two networks it takes about “3 seconds” for MLPs, about “5 seconds” for VGG11 on CIFAR10, and about “7 seconds” for ResNet18 on CIFAR10*. Unfortunately, once again, *falsehoods are being propagated.*
>
> 4. Further, the authors present in Section 5.4 that models trained on disjoint datasets can be merged. However, rather identically in [3], a page-length Section 5.1 demonstrates the precise fact! — as also noted by Reviewer qnqB. ***But as has been a common pattern, there is no mention of this and let alone any comparisons performed.***
> \
> &nbsp;
> ---
>
> \
> &nbsp;
> ### (D) Futility of ‘citing’ related work
> \
> &nbsp;Again, it is obviously not an issue to build upon prior work, **but not presenting them in the correct light or usurping them, as in the glib statements in paragraph 3 of page 9 here, is unacceptable.**
>
> Concretely:
>
> - References [1, 2] are not cited.
> - Then [3] is not “merging models by soft-aligning associations weights” but expressly states that “we primarily utilize exact OT solvers” and thus they obtain exact permutation matrices (see the accuracy of aligned networks before fusion in Table S4 of [3]). We have already seen the tremendous similarity with [3] in the sections (A, B, C) above.
> - Further, the discussion of [4] is strangely expressed as it is not clear why should it be any problem for them to fuse equal-sized networks when also illustrated in their work [4].
> - Lastly [5] is briefly mentioned but the extent of their thematic similarity in regard to mode-connectivity or algorithmic similarity, as also pointed out by Reviewer qnqB, is ignored.
> \
> &nbsp;
> ---
> \
> &nbsp;
> ### Conclusion:
>  \
> &nbsp;Now that we have looked underneath the *façade of Git Re-Basin*, it is crystal clear that in its current form it is but a rehash of prior work [1-5], containing identical algorithms fashioned anew by demonstrating marginal results on additional network sizes and datasets (i.e., ImageNet - where the method fails; results on MNIST, CIFAR10, CIFAR100 were already shown in [2-6])
>
> On that point, we are grateful that ICLR is one of the singular conferences that allow the community to engage in the review process, thereby avoiding inaccurate judgements to seep through in the literature — which in other conferences must only be corrected retrospectively if they are done at all. Given that ICLR is a torchbearer of the modern scientific process and will safeguard scientific integrity, we hope that we can continue to remain confident in ICLR’s righteous and rigorous decision process.
> \
> &nbsp;
>
> Finally, it goes without saying that we are happy to elaborate on or answer any further comments.
>
>
> *Sidak Pal Singh & Martin Jaggi*
>
> ---
> \
> &nbsp;
> **References:**
> 1. Ashmore, Stephen, & Michael Gashler. "A method for finding similarity between multi-layer perceptrons by Forward Bipartite Alignment." 2015 International Joint Conference on Neural Networks (IJCNN). IEEE, 2015.
> 2. Yurochkin, Mikhail, et al. "Bayesian nonparametric federated learning of neural networks." International Conference on Machine Learning. PMLR, 2019.
> 3. Singh, Sidak Pal, & Martin Jaggi. "Model fusion via optimal transport." Advances in Neural Information Processing Systems 33 (2020): 22045-22055.
> 4. Wang, Hongyi, et al. "Federated learning with matched averaging." arXiv preprint arXiv:2002.06440 (2020).
> 5. Tatro, Norman, et al. "Optimizing mode connectivity via neuron alignment." Advances in Neural Information Processing Systems 33 (2020): 15300-15311.
> 6. Entezari, Rahim, et al. "The role of permutation invariance in linear mode connectivity of neural networks." arXiv preprint arXiv:2110.06296 (2021).

---

> > ### Author Response · Authors · 2022-11-12
> > **Git Re-Basin is distinct from (and outperforms!) prior work**
> >
> > We thank the commenters for their interest in our work and for drawing our attention to their prior work. To address the questions raised,
> >
> > # Connections with Activation Matching
> > We are pleasantly surprised to learn of the connections between our method introduced in Sec. 3.1 and the techniques of [3], due to [3] claiming to do soft (i.e. not permutation) matching between units via optimal transport:
> > 1. Abstract of [3]: “We present a layer-wise model fusion algorithm [...] that utilizes optimal transport to **(soft-) align** neurons across the models”
> > 2. Sec. 4 of [3]: “But in practice, it is more likely to have **soft correspondences** between the neurons of the two models [...] This is where optimal transport comes in and provides us a **soft-alignment** matrix in the form of the transport map T.”
> > 3. Conclusion of [3]: “We show that averaging the weights of models, by first doing a layer-wise **(soft) alignment** of the neurons via optimal transport [...]”
> >
> > Additionally, as evidenced by the length of their derivation above, the connection to their work is not obvious. However, after inspecting the code release for [3], we can confirm that [3] does appear to be doing hard permutation matching instead of soft matching as claimed in the text, so our method does indeed appear similar. Despite the similarity, we believe our derivation of hard activation matching from a framework of least-squares regression provides additional motivation for the technique of [3, 5].
> >
> > Note, also, that we found activation matching to be the weakest of the three algorithms explored in our paper. Furthermore, data-privacy concerns can make activation matching untenable, eg. in federated learning.
> >
> > We have edited the paper to highlight the connection between the two techniques.
> >
> > # Claimed Connections with the Weight Matching Method
> > Our weight matching method (Sec. 3.2) differs from prior work in a number of respects:
> > 1. Our method is derived from an objective jointly defined over all layers, and considers the weights in _all_ relevant layers – both previous and following – when selecting new permutations, repeating the process to convergence. In contrast, Algorithm 1 of [3] only completes a single, greedy pass over the layers and only considers the weights of the immediately previous layer when selecting permutations. To their credit, [3] acknowledges this as a shortcoming of their work: “In principle, we should actually search over the space of permutation matrices, jointly across all the layers.” (Sec 4.1 of [3]).
> > 2. Our theoretical contributions, Lemmas 1 and 2, have no parallel in prior work [1-6].
> > 3. Our implementation works on models of nearly any architecture, including skip connections, attention, and so forth. Prior works tend to focus on simpler architectures without skip connections. According to the paper [3] and code, the “wts” algorithm of [3] is only applied to MLPs and CNNs with no bias terms or normalization layers. Note that the ResNet results of [3] are limited to activation matching.
> >
> > In response to this question, we developed a simple example that highlights a failure mode of greedy, single-pass matching in Appendix A.7. In this example, we show that greedy single-pass matching fails to merge two simple identity-function networks, while our weight matching algorithm succeeds.
> >
> > In experiments, we have found our weight matching method to be superior to the greedy single-pass method of [1, 3, 4] in merged model performance. For example, the VGG-11/CIFAR-10 experiment of [3] gives 85.98% test accuracy, whereas we achieve 86.57% and run 4.5x faster, merging their model weights. On ImageNet with ResNet50 models the commenter's method from **[3] gets 1.38%** top-1 accuracy, whereas **we achieve 51.01%**. See Appendix A.7.1 for more info!
> >
> > # Claimed Connections with the Straight-through Estimator Method
> > We fail to see the purported connection. Our straight-through estimator (STE) uses temporary weights to search for a permutation with low barrier. These temporary weights do not participate in the construction of the merged model, beyond identifying a permutation. All of our alignment methods are evaluated in the same way (permute → linearly interpolate → evaluate).
> >
> > Our results are therefore incomparable to the cited results of [3] where the network is further trained **after interpolation**, allowing the model to depart from the line connecting the two models. At no point in any of our experiments do we perform training of model weights post-merge, since doing so should obviously improve model performance.
> >
> > In short: we only interpolate. The method in the commenter’s paper, [3], interpolates and then requires continued training.

---

> > > ### Author Response · Authors · 2022-11-12
> > > **Git Re-Basin is distinct from (and outperforms!) prior work (con't)**
> > >
> > > # Claimed Prior CIFAR-10 Results
> > > This is a false comparison. As we discuss in the previous section, the cited results of [3] rely on continued training of the model after merging. Therefore, they do not demonstrate linear mode connectivity (LMC). In contrast, our method achieves LMC between permutations of model units without requiring any continued training of model weights post-merging.
> > >
> > > To the best of our knowledge, our result is the first such LMC demonstration for ResNets trained on non-trivial datasets.
> > >
> > > # Computation Time
> > > The emphasis of our claim is that we are first to achieve zero-barrier LMC for ResNets. Our remark regarding the running time exists only to highlight a desirable feature of our method. We do not claim to be the first method to run in seconds.
> > >
> > > Nevertheless, we performed a head-to-head comparison against one of [3]’s own experiments. The weight matching algorithm of [3] merged two modified VGG-11 models in 2.86s, achieving a merged accuracy of 85.98%. Our weight matching algorithm performs the same task in 0.64s despite making 6 passes over the layers and achieves a merged accuracy of 86.57%.
> > >
> > > In short: our implementation is 4.5x faster and achieves better results when run against [3]’s own experiments.
> > >
> > > # Merging Models Trained on Disjoint Datasets
> > > We emphasize that at no point have we claimed to be the first to merge models trained on separate datasets. In fact, we open Sec. 5.4 by citing the prior work that inspired this particular experiment. (We have added a citation to [3] in Sec 5.4.) Rather, we present disjoint dataset merging results as an application demonstrating our method’s ability to combine independent and biased models.
> > >
> > > The results in Sec. 5.1 of [3] are limited to MLP models trained on MNIST. In contrast, we show successful experiments with large ResNets trained on CIFAR-100.
> > >
> > > Furthermore, our calibration analysis (Fig. 10) of these merged models has no parallel in [3].
> > >
> > > # Conclusion
> > > To summarize:
> > > 1. We focus on linear mode connectivity modulo permutation symmetries and its implications for a single-basin theory. On the other hand, [3] emphasizes “soft” matching of units via optimal transport.
> > > 2. Our weight matching and straight-through estimator methods solve a problem that the commenter’s own prior work acknowledges as important: “In principle, we should actually search over the space of permutation matrices, jointly across all the layers.” (Sec 4.1 of [3]).
> > > 3. Our weight matching method outperforms prior work in both speed and merged model performance. We achieve 51.01% top-1 accuracy on ImageNet compared with 1.38% using [3].
> > > 4. Claims regarding CIFAR-10 results and our straight-through estimator method are based on false comparisons.
> > >
> > > As a final note, we would like to gently call into question whether accusations of misinformation are conducive to an impartial, factual scientific discourse.

---

> > > ### Public Comment · ~Sidak_Pal_Singh1 · 2022-11-17
> > > **Concerns not addressed yet (3/3)**
> > >
> > > \
> > > &nbsp;
> > > ## C.  Other inconsistencies present in the response and current paper
> > > \
> > > &nbsp;
> > >
> > > 1. **Merging models trained on separate datasets:** A similar setup was already considered in Section 5.1 of [3], as also pointed out by Reviewer qnqB, ​​ and that also obtained very similar results. Indeed, we agree with what the authors say, “The results in Sec. 5.1 of [3] are limited to MLP models trained on MNIST. In contrast, we show successful experiments with large ResNets trained on CIFAR-100.”. Given *the significant extent of similarity in the experimental setups and observations*, we ask that authors *state the same fact above in the main text of their paper and portray the accurate picture*. Presently, [3] is cited superficially on this aspect and under the incorrect name of ‘model patching’. If time permits of course best would be to add [3] as a baseline. \
> > > &nbsp;
> > >
> > > 2. **Results with STE are comparable to methods with retraining.** The authors mention in the paper that their STE-based algorithm allows for learning a better permutation *“using a conventional training loop”* by “initializing to the weight matching solution of Section 3.2 and leveraging the data distribution”. As far as the task of merging models is concerned, this is still comparable to the prior work practice of also *“using a conventional training loop”* by retraining/finetuning from the previous matching solution. Despite these strategies having interesting differences, they are similar enough to merit a connection for they similarly improve their respective solutions by leveraging the data distribution.\
> > > &nbsp;
> > >
> > > 3. **Discussion of comparisons on prior CIFAR10 results:** We would like to ask the authors to also mention that [3] **also shows results without finetuning** and that ‘continued training’ results are only further improvements. First, we already saw in point B.2 above that the two methods have *a mere 0.07% difference in test accuracy for VGG11 on CIFAR10*. If [3] does not demonstrate LMC on CIFAR10, then surely a 0.07% improvement does not qualify for their claims to LMC. Next, even in the ResNet18 case on CIFAR10 and without any retraining, *[3] shows a 58.5% decrease in the LMC barrier* relative to naive interpolation. The authors do not compare the barrier reduction in this setting, which might well be better but is not clear as a priori.\
> > > &nbsp;
> > >
> > > 4. **Past work on linear mode connectivity along the training evolution and increasing width:** Although we highlighted in our previous comment that past work [3] had observed similar effects of model width (Table S11) and training time (see Fig S2, Table S1) to LMC, *the authors have not answered this aspect yet in their response*. To their defense, they did run this on CIFAR10 as compared to just MNIST in [3]. However, we believe that the prior work should still be mentioned when presenting the additional results.\
> > > &nbsp;
> > >
> > > 5. **Evidence for fusing multiple models on different datasets/models:** The authors claim that their method “has been shown to work with as many as 32 models at a time”, while [3] “only demonstrates results on at most 8 models at a time”. However, unlike in other places where they do not fail to emphasize if it is a bigger network/dataset setting, here it is not mentioned that their setting is restricted to MLPs on MNIST *while [3] showed the feat of fusing multiple models on VGG11 and CIFAR100*. We hope it is not too much to ask that the authors mention this in the main paper as well.\
> > > &nbsp;
> > >
> > > ----
> > > \
> > > &nbsp;
> > >
> > > ## Conclusion
> > > \
> > > &nbsp;
> > > All we are asking the authors to do is to (a) acknowledge the similarities and convey them accurately in the main text of the paper; (b) add the baselines from prior work [1-4] specifically in Figures 2, 4 and 5, as proper scientific rigour would demand. (c) acknowledge that linear mode connectivity for both hard and soft alignment was studied in previous work.
> > > This will let the readers truly understand the potential benefits of the proposed algorithms over the prior work, and would help clarify the more than welcome progress after the three years.
> > > \
> > > &nbsp;
> > >
> > >
> > > *Sidak Pal Singh & Martin Jaggi*

---

> > > > ### Author Response · Authors · 2022-11-19
> > > > **Commenter's prior work does not demonstrate LMC on CIFAR-10, and other misconceptions**
> > > >
> > > > # Activation matching
> > > > We clearly recognize prior work in Sec. 3.1: “Activation matching has previously been studied for model merging in Tatro et al. (2020); Singh & Jaggi (2020); Li et al. (2016) albeit not from the perspective of OLS regression.”
> > > >
> > > > [Collier and Dalalyan] study minimax rates for various permutation estimators. They do not study model merging.
> > > >
> > > > Figures 2, S6, S7, S8 of [3] are for the alignment of MLPs on MNIST. In contrast, we demonstrate zero-barrier linear mode connectivity between large ResNet models trained on CIFAR-10.
> > > >
> > > > Again, activation matching was found to be the weakest of the 3 methods under consideration in our work.
> > > >
> > > > # Weight matching
> > > > **Bias terms and other features not supported by [3].** This code snippet is cherry picked. Following the ResNet20 specification, we do not use bias terms in Conv layers in that particular model. However, they are present in the fully-connected layers, as well as in the VGG-16 and MLP architectures. Our ResNet50 experiments are based on off-the-shelf architectures and publicly available weights.
> > > >
> > > > **Computation time.** Careful readers will note that we compare performance of implementations, not their runtime complexities.
> > > >
> > > > **Changing the counterexample.** Sure, obviously changing details of the counterexample breaks it. The counterexample could reasonably be extended to other norms, including L2, leveraging the same general technique.
> > > >
> > > > Furthermore, we show in Lemma 1 that no polytime algorithm can exist producing uniformly good approximations to this problem.
> > > >
> > > > **ResNet50/ImageNet results.** Yes, we recalculated BatchNorm statistics after running OT-Fusion “wts” for experimental parity with our weight matching (Alg. 1).
> > > >
> > > > Performance differences could reasonably be attributed to a number of factors: the presence of skip connections, the vastly greater depth of ResNet50 relative to VGG-11, the lack of normalization layers in [3]’s implementation of VGG-11, and so forth.
> > > >
> > > > In addition, we found that the number of passes over the network layers to be of unique importance. [3] is inherently limited to a single forward pass through the layers of the network. For comparison, if our weight matching algorithm is artificially handicapped to a single pass over the layers, we achieve a similarly low ~7%.
> > > >
> > > > # Straight-through estimator (STE)
> > > > Again, this is a misrepresentation of the straight-through estimator (STE) method (Sec. 3.3). At no point do we ever continue training the model weights after merging.
> > > >
> > > > [3]’s continued training of models post-merge results in a 16.78% percentage point increase in accuracy (77.00% → 93.78%) on ResNet18/CIFAR-10 (Table 1 of [3]). In contrast, we achieve 94.37% **without requiring any additional training.**
> > > >
> > > > Results of our STE optimization can be used to confirm the conjecture of [Entezari et al, 2021]. The cited results with continued training cannot.
> > > >
> > > > # Linear mode connectivity on CIFAR-10
> > > > The commenter’s prior work, [3], does not demonstrate linear mode connectivity on CIFAR-10. Their ResNet18 experiment suffers a 16.2% drop in accuracy (93.2% → 77.0%) after merging (Table 1 of [3]). Their VGG-11 models suffers a drop from 90.5% → 85.98%, hardly a zero-barrier interpolation and well below the [accepted success threshold of 94% on CIFAR-10](https://dawn.cs.stanford.edu/benchmark/#cifar10-train-time).
> > > >
> > > > Our work remains the first to demonstrate zero-barrier linear mode connectivity of large ResNet models trained on CIFAR-10, to the best of our knowledge.

---

> > > ### Public Comment · ~Sidak_Pal_Singh1 · 2022-11-17
> > > **Concerns not addressed yet (2/3)**
> > >
> > > 2. **‘Outperforms’ prior work on CIFAR10:** Next, it is quite starkly visible from one of the comparisons they provide with [3], that the algorithm performs almost the same as the weight-based alignment of [3]. Concretely, on VGG11/CIFAR10 where naive averaging gives 17.02% test accuracy, the authors achieve 86.57% while [3] attains 85.98%. This rather *small difference of 0.6% is already pointing towards an affirmation of our claims about the similarity with the weights-based alignment* in prior work. As a matter of fact, even this 0.6% can be basically removed by proper tuning of the baseline, i.e., by getting rid of the `--normalize-wts` flag in the command listed on GitHub, [3] obtains 86.51%. *Therefore, the difference between [3] and the weight matching of this paper amounts to a mere 0.07%, which is probably not statistically significant* (given these numbers are from single runs). \
> > > &nbsp;
> > >
> > >  3. **‘Outperforms’ prior work in computation time:** About ‘outperforming’ [3] in this VGG11/CIFAR10 case, the authors show that their method takes 0.64s while [3] takes 2.86s. However, one must look thoroughly into this and not be misled by such numbers. First and foremost, the runtime complexity of their algorithm is $\mathcal{O}(T  L n^3)$, while that of exact OTFusion is $\mathcal{O}(L n^3)$, where $n$ is the layer-width, $L$ is the network depth and $T$ is the number of iteration they need for ‘Permutation Coordinate Descent’. Therefore, **OTFusion has lower runtime complexity than the weights-based algorithm here** by a linear factor of $T$. But, then it seems strange that the picture is reversed in practice. Perhaps this can be reconciled by realizing that this is *not a one-to-one comparison*, as the two methods use inherently different frameworks: namely, JAX by Git Re-Basin, and PyTorch by OTFusion, and where the authors rely on JIT-ing (Just In Time compilation) of their code. Thus, this probably boils down to *which code has been more heavily optimized*, and does not point to the better efficiency of their algorithm. \
> > > &nbsp;
> > >
> > >  4. **Fixing the toy example.** For the case of the three-layer neural network of width 2 with linear activations, the authors indeed present an interesting and novel toy example that shows the limitation of greedy weight-based alignment strategies — although, of course, it is hard to say how realistic is the toy example. Nevertheless, there's a simple fix: use Euclidean distance as the ground cost instead of the squared Euclidean (already experimented in [3] via the flag `--not-squared`).
> > >
> > >     While for the squared Euclidean setting the cost matrix (for the second layer) is $\bf{C} = \left[\begin{array}{ll}1 & 2 \\\ \epsilon^2 & (1-\epsilon)^2\end{array}\right]$, in the Euclidean setting $\bf{C} = \left[\begin{array}{ll}1 & \sqrt{2} \\\ \epsilon & 1-\epsilon \end{array}\right]$. Therefore, in the latter, while $\boldsymbol{P}_1 =\bf{I}$ is identity, $\boldsymbol{P}_2=\left[\begin{array}{ll}0 & 1 \\\ 1 & 0\end{array}\right]$ since $\sqrt{2} + \epsilon < 2 - \epsilon$ for small $\epsilon$. Substituting this choice of permutation matrices in their relevant expressions shows that greedy matching recovers exactly the same solution as their weight matching method.
> > > \
> > > &nbsp;
> > >
> > >  5.   **ImageNet results, an abrupt change?:** Further, the authors also present results on ImageNet where surprisingly there seems to be a significant gap between their performance and that of [3], as much as 50%. This seems surprising, *given the two methods hardly differed by 0.6% on CIFAR10*. It isn't clear how the authors ran the OTFusion code: *whether they recalculate the batch-norm statistics as they do for their own method, and whether they cared to tune the hyperparameters* (such as the # of samples for activations) or not (which wouldn't be surprising given point B above ^^ where they did not tune). In fact, *(Anonymous 2023) which the authors cite, makes the precise point that recalculating the batch-norm statistics can result in a drastic improvement*, c.f., Fig 6a of Anonymous 2023 and Fig 8a here. All of this makes the abrupt change in their performances impalpable and a bit dubious, to say the least.\
> > > &nbsp;
> > >
> > > 6. **Adding baselines is already helping Science:** We can already see how adding baseline comparisons indeed benefits everybody, and importantly, Science as well. We (as a community) have **gone from an algorithm that outperforms the next baseline (naive averaging) by ~70% to one only 0.6% (or actually 0.07%) better** when stronger baselines like OTFusion are considered. We've found a counter-example to the greedy strategy and then identified a simple fix to it as well. We now also have an open research question about why the two methods behave very differently in the two settings. *To conclude, it is clear that proper comparisons to baselines do not downgrade anybody’s work, but rather will only provide new research perspectives and a better understanding of the problem at hand.* \
> > > &nbsp;

---

> > > ### Public Comment · ~Sidak_Pal_Singh1 · 2022-11-17
> > > **Concerns not addressed yet (1/3)**
> > >
> > > We thank the authors for their response. We remain nevertheless a bit disappointed by the downplay of crucial aspects of novelty, especially in the main paper. Specifically:\
> > > &nbsp;
> > >
> > >  * (A)  The authors, in the main paper, still *fail to acknowledge the equivalence of the algorithms* in a the activation-based alignment case, and *don’t mention that linear mode connectivity for hard alignment has been shown before*.
> > >
> > > * (B) Baselines and the similarities of the weight-based alignment are still not discussed accurately, *while one of their own experiments now suggests a difference of a **mere 0.6%** in test accuracy* (and *even that 0.6% can be removed* as detailed below).
> > >
> > > * (C) There are other *inconsistencies* present in the response and the paper.
> > > \
> > > &nbsp;
> > > -----
> > > \
> > > &nbsp;
> > > ## (A) Equivalence of the activation-based algorithm
> > >
> > > In the previous comment, we have proven that the first algorithm of the paper is identical to the activations-based OTFusion algorithm when the layer widths are equal. The proof follows directly from the basics of linear programming and also reveals that the permutation search forms the most natural special case of Optimal Transport.
> > >
> > > 1. **Unacknowledged equivalence:** The equivalence is still not acknowledged in the revised main paper.\
> > > &nbsp;
> > >
> > > 2. **Least-squares regression motivation preexists:** In the response, the authors cast their activation matching as providing additional motivation based on least-squares regression. But, thanks to one of the references shared in the public comment by Frederik Benzing, [https://arxiv.org/pdf/1310.4661.pdf, page 10], it can be seen that this motivation is not new and was already explicitly shown in 2013 (see Equations 16 and 17 therein). $\pi^{\mathrm{LSS}} = \arg\min\_{\pi \in \mathfrak{S}\_n} \sum\_{i=1}^n \|X_{\pi(i)}-X_i^{\\#}\|^2 = \arg \min _{\pi \in \mathfrak{S}_n} \operatorname{tr}\left(M P^\pi\right)$  \
> > > &nbsp;
> > >
> > > 3. **’Hard alignment’ is a special case of ‘soft alignment’.** As we have clarified in our previous answer, the permutation case (‘hard alignment’) is the first and most natural special case of the more general optimal transport as considered in [3]. It is thus not accurate to portray [3] as focusing on ‘soft alignment’. [3] will only use soft alignment when networks have unequal layer widths. For instance, when the authors quote [3] as
> > >
> > >    > “But in practice, it is more likely to have soft correspondences between the neurons of the two models [...]”
> > >
> > >     they do not disclose the relevant context [...], which is “for a given layer, **especially if their number is not the same across the two models**”
> > >
> > >    The notion of ‘soft’ alignment in [3] is only mentioned 4 times out of 86 when alignment is being discussed. There are also places where it is expressly stated that the matching can “be formulated as a **permutation matrix** and “just multiplying the parameters by this matrix would align the parameters” and that **exact OT** is used. As mentioned, Figure 1 of [3] clearly shows the hard alignment with permutation matrices. Thus, it is a false argument when the authors claim that [3] “emphasizes “soft” (ie., non-permutation) matching”.
> > > \
> > > &nbsp;
> > > In summary, **Linear mode connectivity for hard alignment has been shown before**, as in Figures 2, S6, S7, S8 and Tables 1, S4 of [3] (see also B.2 and C.2 below). We’d be grateful if the authors would rightfully acknowledge this fact in the main paper.
> > >
> > > ----
> > > \
> > > &nbsp;
> > >
> > > ## B.  Similarities with the weights-based algorithm are not acknowledged
> > >
> > > \
> > > &nbsp;
> > > We already mentioned in our initial comment that, as compared to the prior works [1-4], Git Re-basin matches by weights in an **alternate** — albeit in a highly similar way. Instead of acknowledging the similarities in their response or in updates to Section 3.2, the authors downplay them and further extol the differences. \
> > > &nbsp;
> > >
> > > 1. **‘Handling’ architectural aspects like bias terms:** In particular, in their response, the authors mention that their algorithm can handle architectures with bias terms and normalization layers in contrast to prior work. This gives the wrong impression that there is no prior work on weights-based alignment. Further, as an aside, if we inspect their code, even they too disable bias `use_bias=False` and use `nn.LayerNorm()` instead of BatchNorm:
> > >    ```
> > > self.norm1 = nn.LayerNorm()
> > > self.conv2 = nn.Conv(features=self.num_channels, kernel_size=(3, 3), strides=1, use_bias=False)
> > > self.norm2 = nn.LayerNorm()
> > >    ```
> > > If what they meant to say is that in principle they can handle bias terms and the like, the same can be said about algorithms from prior work too.

---

> ### Public Comment · ~Sidak_Pal_Singh1 · 2022-11-06
> **Git Clone or Git Re-Basin? (2/3)**
>
> There seems to be still one difference: the domain for OT is the set of bistochastic matrices while that of Git Re-Basin is the set of permutation matrices. But, as anyone having taken a course on linear programming would know, the solution to a linear program is found at the extreme points of the polytope (i.e., vertices). Thanks to the well-known Birkhoff-von Neumann theorem, extreme points of the Birkhoff polytope (bistochastic matrices) are precisely the permutation matrices, and thus the solutions to both problems are identical.
>
> Therefore, ***Method 1 of this work is simply a special case of [3], and moreover, identical to [3] when considering their activations-based ground cost.***
>
> *Remark:* The authors remark that as a further variant of their Method 1, it is also possible to use the cross-correlation matrix of activations. It must be noted that this has already been considered in [5, page 6].
>
> Let us invite you to take a glimpse at what Git Re-Basin has to say:
>
> > "Matching methods. We propose **three new algorithms**, grounded in concepts and techniques from combinatorial optimization, to align the weights"
>
> But wait — let's not overlook the other two algorithms, which too have a similar fate!
> \
> &nbsp;
>
> ---
> \
> &nbsp;
> ### (B) The high similarity of Methods 2 and 3 to prior work [1,2,3,4] and the lack of baseline comparisons
> The second 'novel' method that Git Re-Basin proposes is to "inspect the weights of the model itself" in order to align neurons. This is a natural strategy to consider, and in fact, *similar approaches have been employed in numerous past works* --- **going back to 2015 [1] as well as [2,3]**. Likewise, *all these past approaches also do not need to rely on input distributions* (e.g., weights-based alignment of [3]) and can run in seconds (see Section C.3 below), unlike the impression conveyed in Git Re-Basin.
>
> More specifically, as remarked in their work, in contrast to activations-based alignment, weight-based matching inherently poses a more involved bilinear assignment problem which is computationally hard to solve exactly. Most of the prior works ([1-3]) use a greedy layerwise alignment of neurons from the input to the output layer. Git Re-Basin does it in an alternate way which might be slightly better than the above prior works although at the cost of more expensive computations and running time. *However, unfortunately, Git Re-Basin fails to acknowledge this at all and completely sweeps these similarities under the carpet. Let alone showing a comparison to prior works!*
>
> Furthermore, the past works [2-4] opt to use additional computation to fine-tune/retrain after this initial alignment, to instead bridge any downsides. This **brings us to Method 3**, where they use the 'STE estimator', to alternate for a number of iterations between finding an alignment and — guess what — retraining, like for an epoch (what is termed here as the ‘backwards pass’). Now, it should be rather obvious that this strategy is *very similar to the fine-tuning/re-training approach used in [2-4]*. In fact, two further concrete points can be mentioned to showcase this:
> \
> &nbsp;
>
>
> 1. When combined with retraining *just a single weight-based alignment at the beginning suffices* without having to find completely new alignments after every step, as shown in [3]. In this light, it would be interesting to see if the authors notice any significant performance degradation (if at all) when they too perform their particular weights-based alignment only once in their STE method.
>
> 2. More surprisingly, when fusing two networks, [3] also find that simply retraining the naive or vanilla average of the networks performs competitively. In other words, **just using identity as an initial permutation matrix when combined with retraining** (or something in the fashion of the STE method) *can be rather competitive*.
>
> However, once again, the authors fail to carry out these comparisons while enveloping their method with an air of novelty. ***Including the baselines from prior work would perhaps question the ‘novelty’, right?***
> \
> &nbsp;
> ---
> \
> &nbsp;
> ### (C) Several results or observations are already known in similar forms
>
> 1. The authors claim that
> >*“we contribute the first-ever demonstration of zero-barrier LMC between two independently trained ResNets”*, *“the first such demonstration to the authors’ knowledge.”*.
>
>     But we would like to point to Table 1 of [3], where for ResNet18 on CIFAR10 it has been shown that OTFusion (with retraining, like in STE) attains an accuracy of `93.78` while the individual networks have an accuracy of `93.11` and `93.20`. This should suggest preliminary evidence for negligible barrier LMC between two independently trained ResNets, as the OTFusion network corresponds to $\lambda=0.5$ in the interpolation curves and which is generally the point of maximum deviation from the performances of the individual networks (as also seen in Fig 2 here).
>
>  *(continued)*

---

> ### Public Comment · ~Frederik_Benzing1 · 2022-11-12
> **OG**
>
> It should be noted that OTFusion's ideas had already been described in prior work:
> [1] https://arxiv.org/pdf/1511.07543.pdf (Figure 1 here uses precisely the same algorithm as OTFusion)
> As pointed out somewhere below, the equivalence of the approach from [1] and OTFusion follows from basic results.
>
> Also worth noting, even it the connection might be a little more lose:
> https://arxiv.org/pdf/1310.4661.pdf

---

> > ### Public Comment · ~Sidak_Pal_Singh1 · 2022-11-17
> > **Would you mind elaborating?**
> >
> > Dear Frederik,
> >
> > Thanks for your comment and for sharing the two references.
> >
> > - First, [1,https://arxiv.org/pdf/1511.07543.pdf] measures the alignment of feature maps of various layers using their activations and bipartite matching (or semi-matching). Naturally, we cite them in the opening line of the related-work section titled alignment-based methods in our paper, as “ Alignment of neurons was considered in Li et al. [17] to probe the representations learned by different networks.” However, **it should also be noted that** [1] *aims to “propose a specific method of probing representations: training multiple networks and then comparing and contrasting their individual, learned representations at the level of neurons or groups of neurons.”*, and claims to have *"demonstrated a method for **quantifying the feature similarity** between different, independently trained deep neural networks."*.  \
> > &nbsp;
> >
> >
> >
> > - Then we would be grateful **if you could please point us and the readers to where [1] uses their alignment strategy to align an entire network and then merge with another network**. Let us even put aside if they also use other strategies that we use, like weight-based alignment, or if they fuse with an unequal-sized network, or multiple same-sized networks. Likewise, let us put aside fact whether [1] manages to pull this off successfully or not. \
> > &nbsp;
> >
> > *Sidak Pal Singh*

---

### Public Comment · ~Keller_Jordan1 · 2022-11-18
**error: Please commit your changes before merging.**

Git Re-Basin presents exciting new results: zero-barrier mode connectivity for ResNet20s on CIFAR-10, and an improvement of the test set performance of merged ResNet50s on ImageNet from ~1% to ~50%.

The authors argue that their results are due to several novel neuron-alignment algorithms, including a new and efficient weight-matching scheme. Other commenters have argued that some of these may be special cases of previously designed algorithms, but we do not focus on this point.

Instead, **we claim that the new results of Git Re-Basin are essentially the product of two implementation details**, both of which relate to *controlling the internal statistics of the interpolated networks* [3]:

- For **CIFAR-10:** The authors use ResNets in which the standard BatchNorm layers have been swapped out for LayerNorm.
- For **ImageNet:** The authors reset BatchNorm layer statistics after merging.

In particular, [3] finds that if the above details are taken into account, then the baseline neuron-alignment method of [1] (a highly-cited work that dates back to 2015) is sufficient to accomplish the results of Git Re-Basin. Whether the strong OTfusion method of [2] could further improve on these results remains a plausible and interesting open question.

We believe that the main text of the paper, as it currently stands, is unclear. Improvements to existing neuron-alignment algorithms, which do not appear to be necessary for such results, are emphasized. And the two essential changes above are relegated to the appendix, rather than committed to the main text. We feel that the relative importance of these different factors should be made clear.

Thank you for your time.

[1] Li, Yixuan, et al. "Convergent learning: Do different neural networks learn the same representations?." arXiv preprint arXiv:1511.07543 (2015).

[2] Singh, Sidak Pal, and Martin Jaggi. "Model fusion via optimal transport." Advances in Neural Information Processing Systems 33 (2020): 22045-22055.

[3] Jordan, Keller, et al. "REPAIR: REnormalizing Permuted Activations for Interpolation Repair." arXiv preprint arXiv:2211.08403 (2022).

---

> ### Author Response · Authors · 2022-11-19
> **Our experiments controlled for that. Also, REPAIR is follow-up work to Git Re-Basin.**
>
> 1. We emphasize that the alignment of units in networks (Git Re-Basin) and the tuning of normalization layers after interpolation (REPAIR, [3]) are orthogonal concerns and improvements in either are welcome. In fact, one could apply REPAIR to the matchings output by Git Re-Basin, potentially giving better results than reported in either paper.
>
> 2. Claims that our improvements are due to the recalculation of BatchNorm statistics are incorrect. We controlled for this difference by recalculating BatchNorm statistics after merging for _every_ ResNet50/ImageNet result we report, a common technique when interpolating weights [Izmailov et al, 2018; Wortsman et al, 2021; Maddox et al, 2019; Wang et al, 2021]. To clarify: in the ResNet50/ImageNet experiment, we compare our weight matching algorithm against [2]’s “wts” algorithm, not the activation matching method as is applied in [3].
>
> 3. We highlight that our weight matching algorithm does not require knowledge of the data or loss, a crucial advantage over the data-aware matching algorithm used in [3] when considering applications such as federated learning. Notably, in many settings we achieve similar performance to the techniques presented in [3] *despite having no access to the training data.*
>
> 4. Arguments that the novelty of our work is diminished by the commenter’s work, [3], are peculiar considering that [3] extensively cites Git Re-Basin as prior work. Furthermore, the commenter’s paper, [3], was submitted to arXiv a mere 3 days ago.

---

### Public Comment · ~Seok-Ju_Hahn1 · 2023-02-09
**Requests on missing citation related to Federated Learning and LMC**

Dear authors,

Hi, I am a PhD student researching on the federated learning algorithm.
First, I would like to deeply congratulate you on the acceptance of this work, `Git Re-Basin` with great scores!

Though I have mailed to the first author several times, I couldn't get replies from him, so I posted this comment here in OpenReview.

As discussed in section 5.4 of the paper, inducing LMC by combining separately training models on the disjoint data can surely be extended to the federated learning, as did in [a prior work](https://dl.acm.org/doi/abs/10.1145/3534678.3539254) [1], which focused on enhancing personalization peformances by inducing LMC between globally communicated model (i.e., federated models) and several local personalized models trained on statistically heterogeneous datasets.

To the best of my knowledge, [1] was **the first to introduce LMC to federated learning** for a better personalization performance. In this work, we have induced LMC with more severe `dataset disjointedness` scenario (i.e., *statistical heterogeneity*, or *non-IIDness* in FL community).

Interestingly, it was observed that the global model can be successfully connected (i.e., LMC is induced) to each different local personalized model during federated training, thus have wider minima (please see Figure A1 in p.11) with other benefits; better personalization performances, robustness to the label noise, and broader applicability including language modeling (i.e., LMC is also induced between two differently initialized LSTM models).

Although authors of `Git Re-Basin` have mentioned and cited some works related to federated learning in section 6 of the paper, I think my work [1] was somewhat unfortunate to be cited.

I would like to kindly request to check these concerns and consider citing this work related to *federated learning* and *LMC* in your paper of camera-ready version?
I am looking forward to receiving your positive response.
Thank you.

Best,
Seok-Ju Hahn

### Reference
* [1] Hahn, S. J., Jeong, M., & Lee, J. (2022, August). Connecting Low-Loss Subspace for Personalized Federated Learning. In Proceedings of the 28th ACM SIGKDD Conference on Knowledge Discovery and Data Mining (pp. 505-515). (https://dl.acm.org/doi/abs/10.1145/3534678.3539254)

---

### Decision · Program_Chairs · 2023-01-20

**Decision:**

Accept: notable-top-5%

**Justification For Why Not Higher Score:**

NA

**Justification For Why Not Lower Score:**

The paper does make useful observations about the question of basins and permutation symmetries, and these would be of interest to the community. Towards the talk, the authors are encouraged to clearly explain their contribution in the context of prior work on this problem.

**Metareview: Summary, Strengths And Weaknesses:**

(a) This paper addresses an important problem of understanding the relation between different learned models trained on the same training data (due to different initialization and SGD batches). Prior work had conjectured that these models are in fact "the same up to permutation". The current paper provides further support of this by using a matching method that aligns different multilayer models.
(b) The main interesting finding of this paper is linear-mode-connectivity (ie all solutions are the same up to permutation) for CIFAR datasets, as opposed to MNIST shown before.
(c) The paper suffers to some extent from over-claiming and lack of rigor (e.g., the author call a "counter-example" something which is not clearly a counter-example). The matching algorithm used is quite straight-forward, and arguably equivalent to transport-based methods. So while there is a real contribution here, the authors could have delivered it in a more scientific manner.

**Note From Pc:**

if the above contains the word "oral" or "spotlight" please see: "oral" presentation means -> notable-top-5% and "spotlight" means -> notable-top-25%. As stated in our emails, we are disassociating presentation type from AC recommendations